# AdaFed: Fair Federated Learning via Adaptive Common Descent Direction

**Shayan Mohajer Hamidi**                                       *smohajer@uwaterloo.ca*
*Department of Electrical and Computer Engineering*
*University of Waterloo*

**En-Hui Yang**                                                 *ehyang@uwaterloo.ca*
*Department of Electrical and Computer Engineering*
*University of Waterloo*

**Reviewed on OpenReview:** *https://openreview.net/forum?id=rFecyFpFUp*

## Abstract

Federated learning (FL) is a promising technology via which some edge devices/clients collaboratively train a machine learning model orchestrated by a server. Learning an unfair model is known as a critical problem in federated learning, where the trained model may unfairly advantage or disadvantage some of the devices. To tackle this problem, in this work, we propose AdaFed. The goal of AdaFed is to find an updating direction for the server along which (i) all the clients' loss functions are decreasing; and (ii) more importantly, the loss functions for the clients with larger values decrease with a higher rate. AdaFed adaptively tunes this common direction based on the values of local gradients and loss functions. We validate the effectiveness of AdaFed on a suite of federated datasets, and demonstrate that AdaFed outperforms state-of-the-art fair FL methods.

## 1 Introduction

Conventionally, a machine learning (ML) model is trained in a centralized approach where the training data is available at a data center or a cloud server. However, in many new applications, devices often do not want to share their private data with a remote server. As a remedy, federated learning (FL) was proposed in McMahan et al. (2017) where each device participates in training using only locally available dataset with the help of a server. Specifically, in FL, devices share only their local updates with the server, and not their raw dataset A well-known setup to carry out such decentralized training is FedAvg McMahan et al. (2017) which combines local stochastic gradient descent (SGD) on each client with iterative model averaging. The server sends the most recent global model to some selected clients (Eichner et al., 2019; Wang et al., 2021a), and then these clients perform a number of epochs of local SGD on their local training data and send the local gradients back to the central server. The server then finds the (weighted) average of the gradients to update the global model, and the process repeats.

In FedAvg, the vector of *averaged gradients* computed by the server is in fact a common direction along which the global model is updated. However, finding the common direction in this manner may result in a direction which is not descent for some clients. Consequently, the learnt model could perform quite poorly once applied to the private dataset of the clients, yielding an unfair global model (Li et al., 2019a; Bonawitz et al., 2019; Kairouz et al., 2021); that is, although the average accuracy might be high, some clients whose data distributions differ from the majority of the clients are prone to perform poorly on the learnt model.

One possible method to find a direction that is descent for all the clients is to treat the FL task as a multi-objective minimization (MoM) problem Hu et al. (2022). In this setup, a Pareto-stationary solution of the MoM yields a descent direction for all the clients. However, having a common descent direction is not

enough *per se* to train a fair model with uniform test accuracies across the clients[1]. This is because data heterogeneity across different clients makes the local loss functions vary significantly in values, and therefore those loss functions with larger values should decrease with a higher rate to learn a fair model.

To address the above-mentioned issues and to train a fair global model, in this work, we propose AdaFed. The aim of AdaFed is to help the sever to find a common direction $\mathfrak{d}_t$ (i) that is descent for *all* the clients, which is a *necessary* condition to decrease the clients' loss functions in the SGD algorithm; and (ii) along which the loss functions with larger values decrease with higher rates. The latter is to enforce obtaining a global model with uniform test accuracies across the clients.

We note that if the directional derivatives of clients' loss functions along the normalized common direction $\mathfrak{d}_t$ are all positive, then $-\mathfrak{d}$ is a common descent direction for all the clients. As such, AdaFed adaptively tunes $\mathfrak{d}_t$ such that these directional derivatives (i) remain positive over the course of FL process, and (ii) are larger for loss functions with higher values enforcing them to decrease more during the global updation by the server.

The contributions of the paper are summarized as follows:

- We introduce AdaFed, a method to realize fair FL via adaptive common descent direction.

- We provide a closed-form solution for the common direction found by AdaFed. This is in contrast with many existing fair FL methods which deploy iterative or generic quadratic programming methods.

- Under some common assumptions in FL literature, we prove the convergence of AdaFed under different FL setups to a Pareto-stationary solution.

- By conducting thorough experiments over seven different datasets (six vision datasets, and a language one), we show that AdaFed can yield a higher level of fairness among the clients while achieving similar prediction accuracy compared to the state-of-the-art fair FL algorithms.

- The experiments conducted in this paper evaluate many existing fair FL algorithms over different datasets under different FL setups, and therefore can pave the way for future researches.

## 2 Related Works

There are many different perspectives in the literature to combat the problem of fairness in FL. These methods include client selection (Nishio & Yonetani, 2019; Huang et al., 2020a; 2022; Yang et al., 2021), contribution Evaluation (Zhang et al., 2020; Lyu et al., 2020; Song et al., 2021; Le et al., 2021), incentive mechanisms (Zhang et al., 2021; Kang et al., 2019; Ye et al., 2020; Zhang et al., 2020), and the methods based on the loss function. Specifically, our work falls into the latter category. In this approach, the goal is to attain uniform performance across the clients in terms of test accuracy. To this end, the works using this approach target to reduce the variance of test accuracy across the participating clients. In the following, we briefly review some of these works.

One of the pioneering methods in this realm is agnostic federated learning (AFL) (Mohri et al., 2019). AFL optimizes the global model for the worse-case realization of weighted combination of the user distributions. Their approach boils down to solving a saddle-point optimization problem for which they used a fast stochastic optimization algorithm. Yet, AFL performs well only for a small number of clients, and when the size of participating clients becomes large, the generalization guarantees of the model may not be satisfied. Du et al. (2021) deployed the notation of AFL and proposed the AgnosticFair algorithm. Specifically, they linearly parametrized model weights by kernel functions and showed that AFL can be viewed as a special case of AgnosticFair. To overcome the generalization problem in AFL, q-fair federated learning (q-FFL) (Li et al., 2019a) was proposed to achieve more uniform test accuracy across users. The main idea of q-FFL stemmed

---

[1]Similarly to other fields like ML (Barocas et al., 2017), communications (Huaizhou et al., 2013), and justice (Rawls, 2020), the notion of fairness does not have a unique definition in FL. However, following (Li et al., 2019a; 2021), we use standard deviation of the clients' test accuracies—and some other metrics discussed in Section 7—to measure how uniform the global model performs across the clients. Please refer to Appendix C for more in-depth discussions.

from fair resource allocation methods in wireless communication networks. Afterward, Li et al. (2020a) developed TERM, a tilted empirical risk minimization algorithm which handles outliers and class imbalance in statistical estimation procedures. Compared to q-FFL, TERM has demonstrated better performance in many FL applications. Deploying a similar notion, Huang et al. (2020b) proposed using training accuracy and frequency to adjust weights of devices to promote fairness. Furthermore, FCFC Cui et al. (2021) minimizes the loss of the worst-performing client, leading to a version of AFL. Later, Li et al. (2021) devised Ditto, a multitask personalized FL algorithm. After optimizing a global objective function, Ditto allows local devices to run more steps of SGD, subject to some constraints, to minimize their own losses. Ditto can significantly improve testing accuracy among local devices and encourage fairness.

Our approach is more similar to *FedMGDA+* (Hu et al., 2022), which treats the FL task as a multi-objective optimization problem. In this scenario, the goal is to minimize the loss function of each FL client simultaneously. To avoid sacrificing the performance of any client, *FedMGDA+* uses Pareto-stationary solutions to find a common descent direction for all selected clients.

## 3 Notation and Preliminaries

### 3.1 Notation

We denote by $[K]$ the set of integers $\{1, 2, \cdots, K\}$. In addition, we define $\{f_k\}_{k \in [K]} = \{f_1, f_2, \ldots, f_K\}$ for a scalar/function $f$. We use bold-symbol small letters to represent vectors. Denote by $\mathfrak{u}_i$ the $i$-th element of vector $\mathfrak{u}$. For two vectors $\mathfrak{u}, \mathfrak{v} \in \mathbb{R}^d$, we say $\mathfrak{u} \leq \mathfrak{v}$ iff $\mathfrak{u}_i \leq \mathfrak{v}_i$ for $\forall i \in [d]$, i.e., two vectors are compared w.r.t. partial ordering. In addition, denote by $\mathfrak{v} \cdot \mathfrak{u}$ their inner product, and by $\text{proj}_{\mathfrak{u}}(\mathfrak{v}) = \frac{\mathfrak{v} \cdot \mathfrak{u}}{\mathfrak{u} \cdot \mathfrak{u}} \mathfrak{u}$ the projection of $\mathfrak{v}$ onto the line spanned by $\mathfrak{u}$.

### 3.2 Preliminaries and Definitions

In Hu et al. (2022), authors demonstrated that FL can be regarded as multi-objective minimization (MoM) problem. In particular, denote by $\boldsymbol{f}(\boldsymbol{\theta}) = \{f_k(\boldsymbol{\theta})\}_{k \in [K]}$ the set of local clients' objective functions. Then, the aim of MoM is to solve

$$\boldsymbol{\theta}^* = \arg \min_{\boldsymbol{\theta}} \boldsymbol{f}(\boldsymbol{\theta}), \tag{1}$$

where the minimization is performed w.r.t. the *partial ordering*. Finding $\boldsymbol{\theta}^*$ could enforce fairness among the users since by setting setting $\boldsymbol{\theta} = \boldsymbol{\theta}^*$, it is not possible to reduce any of the local objective functions $f_k$ without increasing at least another one. Here, $\boldsymbol{\theta}^*$ is called a Pareto-optimal solution of Equation (1). In addition, the collection of function values $\{f_k(\boldsymbol{\theta}^*)\}_{k \in [K]}$ of all the Pareto points $\boldsymbol{\theta}^*$ is called the Pareto front.

Although finding Pareto-optimal solutions can be challenging, there are several methods to identify the Pareto-stationary solutions instead, which are defined as follows:

**Definition 3.1.** Pareto-stationary (Mukai, 1980): The vector $\boldsymbol{\theta}^*$ is said to be Pareto-stationary iff there exists a convex combination of the gradient-vectors $\{\mathfrak{g}_k(\boldsymbol{\theta}^*)\}_{k \in [K]}$ which is equal to zero; that is, $\sum_{k=1}^{K} \lambda_k \mathfrak{g}_k(\boldsymbol{\theta}^*) = 0$, where $\boldsymbol{\lambda} \geq 0$, and $\sum_{k=1}^{K} \lambda_k = 1$.

**Lemma 3.2.** (Mukai, 1980) Any Pareto-optimal solution is Pareto-stationary. On the other hand, if all $\{f_k(\boldsymbol{\theta})\}_{k \in [K]}$'s are convex, then any Pareto-stationary solution is weakly Pareto optimal [2].

There are many methods in the literature to find Pareto-stationary solutions among which we elaborate on two well-known ones, namely linear scalarization and Multiple gradient descent algorithm (MGDA) (Mukai, 1980; Fliege & Svaiter, 2000; Désidéri, 2012).

---

[2] $\boldsymbol{\theta}^*$ is called a weakly Pareto-optimal solution of Equation (1) if there does not exist any $\boldsymbol{\theta}$ such that $f(\boldsymbol{\theta}) < f(\boldsymbol{\theta}^*)$; meaning that, it is not possible to improve *all* of the objective functions in $f(\boldsymbol{\theta}^*)$. Obviously, any Pareto optimal solution is also weakly Pareto-optimal but the converse may not hold.

• **Linear scalarization:** this approach is essentially the core principle behind the FedAvg algorithm. To elucidate, in FedAvg, the server updates $\boldsymbol{\theta}$ by minimizing the weighted average of clients' loss functions:

$$\min_{\boldsymbol{\theta}} f(\boldsymbol{\theta}) = \sum_{k=1}^{K} \lambda_k f_k(\boldsymbol{\theta}), \tag{2}$$

where the weights $\{\lambda_k\}_{k \in [K]}$ are assigned by the server and satisfy $\sum_{k=1}^{K} \lambda_k = 1$. These fixed $\{\lambda_k\}_{k \in [K]}$ are assigned based on some priori information about the clients such as the size of their datasets. We note that different values for $\{\lambda_k\}_{k \in [K]}$ yield different Pareto-stationary solutions.

Referring to Definition 3.1, any solutions of Equation (2) is a Pareto-stationary solution of Equation (1). To perform FedAvg, at iteration $t$, client $k$, $k \in [K]$ sends its gradient vector $\mathfrak{g}_k(\boldsymbol{\theta}_t)$ to the server, and server updates the global model as

$$\boldsymbol{\theta}_{t+1} = \boldsymbol{\theta}_t - \eta_t \mathfrak{d}_t, \quad \text{where} \quad \mathfrak{d}_t = \sum_{k=1}^{K} \lambda_k \mathfrak{g}_k(\boldsymbol{\theta}_t). \tag{3}$$

However, linear scalarization can only converge to Pareto points that lie on the *convex* envelop of the Pareto front (Boyd & Vandenberghe, 2004). Furthermore, the weighted average of the gradients with pre-defined weights yields a vector $\mathfrak{d}_t$ whose direction might not be descent for all the clients; because some clients may have conflicting gradients with opposing directions due to the heterogeneity of their local datasets (Wang et al., 2021b). As a result, FedAvg may result in an unfair accuracy distribution among the clients (Li et al., 2019a; Mohri et al., 2019).

• **MGDA:** To mitigate the above issue, (Hu et al., 2022) proposed to exploit MGDA algorithm in FL to converge to a fair solution on the Pareto front. Unlike linear scalarization, MGDA adaptively tunes $\{\lambda_k\}_{k \in [K]}$ by finding the minimal-norm element of the convex hull of the gradient vectors defined as follows (we drop the dependence of $\mathfrak{g}_k$ to $\boldsymbol{\theta}_t$ for ease of notation hereafter)

$$\mathcal{G} = \{\boldsymbol{g} \in \mathbb{R}^d | \boldsymbol{g} = \sum_{k=1}^{K} \lambda_k \mathfrak{g}_k; \ \lambda_k \geq 0; \ \sum_{k=1}^{K} \lambda_k = 1\}. \tag{4}$$

Denote the minimal-norm element of $\mathcal{G}$ by $\mathfrak{d}(\mathcal{G})$. Then, either (i) $\mathfrak{d}(\mathcal{G}) = 0$, and therefore based on Lemma 3.2 $\mathfrak{d}(\mathcal{G})$ is a Pareto-stationary point; or (ii) $\mathfrak{d}(\mathcal{G}) \neq 0$ and the direction of $-\mathfrak{d}(\mathcal{G})$ is a common descent direction for all the objective functions $\{f_k(\boldsymbol{\theta})\}_{k \in [K]}$ (Désidéri, 2009), meaning that all the directional derivatives $\{\mathfrak{g}_k \cdot \mathfrak{d}(\mathcal{G})\}_{k \in [K]}$ are positive. Having positive directional derivatives is a *necessary* condition to ensure that the common direction is descent for all the objective functions.

## 4 Motivation and Methodology

We first discuss our motivation in Section 4.1, and then elaborate on the methodology in Section 4.2.

### 4.1 Motivation

Although any solutions on the Pareto front is fair in the sense that decreasing one of the loss functions is not possible without sacrificing some others, not all of such solutions impose uniformity among the loss functions (see Figure 1a). As such, we aim to find solutions on the Pareto front which enjoy such uniformity.

First we note that having a common descent direction is a necessary condition to find such uniform solutions; but not enough. Additionally, we stipulate that the rate of decrease in the loss function should be greater for clients whose loss functions are larger. In fact, the purpose of this paper is to find an updation direction for the server that satisfies both of the following conditions at the same time:

- *Condition (I)*: It is a descent direction for all $\{f_k(\boldsymbol{\theta})\}_{k \in [K]}$, which is a *necessary* condition for the loss functions to decrease when the server updates the global model along that direction.

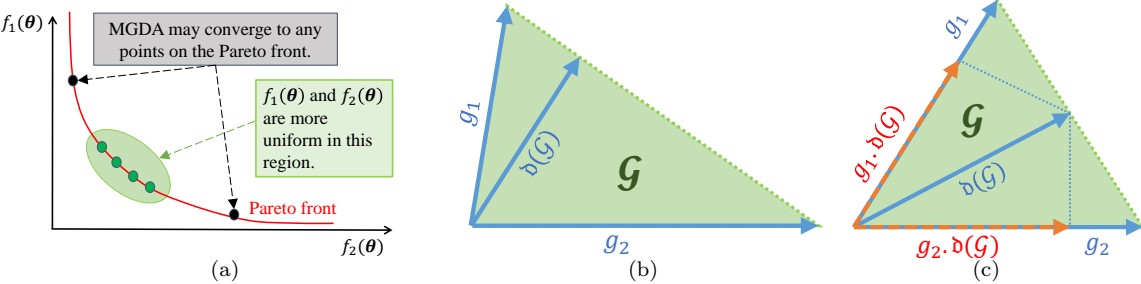

Figure 1: (a) The Pareto front for two objective functions $f_1(\boldsymbol{\theta})$ and $f_2(\boldsymbol{\theta})$ is depicted. MGDA may converge to any points on the Pareto front. (b)-(c) Illustration of convex hull $\mathcal{G}$ and minimal-norm vector $\mathfrak{d}(\mathcal{G})$ for two gradient vectors $\mathfrak{g}_1$ and $\mathfrak{g}_2$. In (b), $\|\mathfrak{g}_1\|_2^2 < \|\mathfrak{g}_2\|_2^2$, where the direction of $\mathfrak{d}(\mathcal{G})$ is more inclined toward $\mathfrak{g}_1$. In (c), $\|\mathfrak{g}_1\|_2^2 = \|\mathfrak{g}_2\|_2^2 = 1$, where the direction of $\mathfrak{d}(\mathcal{G})$ is the same as that of the bisection of $\mathfrak{g}_1$ and $\mathfrak{g}_2$.

- *Condition (II)*: It is more inclined toward the clients with larger losses, and therefore the directional derivatives of loss functions over the common direction are larger for those with larger loss functions.

To satisfy *Condition (I)*, it is enough to find $\mathfrak{d}(\mathcal{G})$ using MGDA algorithm (as Hu et al. (2022) uses MGDA to enforce fairness in FL setup). Nevertheless, we aim to further satisfy *Condition (II)* on top of *Condition (I)*. To this end, we investigate the direction of $\mathfrak{d}(\mathcal{G})$, and note that it is more inclined toward that of $\min\{\|\mathfrak{g}_k\|_2^2\}_{k\in[K]}$. For instance, consider the simple example depicted in Figure 1b, where $\|\mathfrak{g}_1\|_2^2 < \|\mathfrak{g}_2\|_2^2$. The Convex hull $\mathcal{G}$ and the $\mathfrak{d}(\mathcal{G})$ are depicted for $\mathfrak{g}_1$ and $\mathfrak{g}_2$. As seen, the direction of $\mathfrak{d}(\mathcal{G})$ is mostly influenced by that of $\mathfrak{g}_1$.

However, this phenomenon is not favourable for satisfying *Condition (II)* since after some rounds of communication between the server and clients, the value of $\mathfrak{g}$ becomes small for those objective functions which are close to their minimum points. Consequently, the direction of $\mathfrak{d}(\mathcal{G})$ is mostly controlled by these small $\mathfrak{g}$'s, which is undesirable. Note that $\mathfrak{g}_k \cdot \mathfrak{d}(\mathcal{G})$ represents how fast $f_k(\boldsymbol{\theta})$ changes if $\boldsymbol{\theta}$ changes in the direction of $\mathfrak{d}(\mathcal{G})$. In fact, the direction of $\mathfrak{d}(\mathcal{G})$ should be more inclined toward the gradients of those clients with larger loss functions.

One possible solution could be to naively normalize $\{\mathfrak{g}_k\}_{k\in[K]}$ by their norm to obtain $\{\frac{\mathfrak{g}_k}{\|\mathfrak{g}_k\|_2^2}\}_{k\in[K]}$ whose convex hull is denoted by $\mathcal{G}_{\text{norm}}$, and then use this normalized set of gradients to find $\mathfrak{d}(\mathcal{G}_{\text{norm}})$. Yet, the normalization makes all the $\{\mathfrak{g}_k \cdot \mathfrak{d}(\mathcal{G}_{\text{norm}})\}_{k\in[K]}$ equal (see Figure 1c) which is still undesirable as the rate of decrease becomes equal for all $\{f_k(\boldsymbol{\theta})\}_{k\in[K]}$.

Based on these observations, the gradient vectors should be somehow *scaled* if one aims to also satisfy *Condition (II)*. Finding such *scaling* factor is not straight-forward in general. To tackle this issue, and to be able to find a closed-form formula, we find the minimal-norm vector in the convex hull of mutually-orthogonal *scaled* gradients instead, and prove that this yields a common direction for which both *Conditions (I)* and *(II)* are satisfied.

## 4.2 Methodology

To devise an appropriate *scaling* as explained above, we carry out the following two phases.

### 4.2.1 *Phase 1, orthogonalization*

In order to be able to find a closed-form formula for the common descent direction, in the first phase, we orthogonalize the gradients.

Once the gradient updates $\{\mathfrak{g}_k\}_{k\in[K]}$ are transmitted by the clients, the server first generates a mutually orthogonal [3] set $\{\tilde{\mathfrak{g}}_k\}_{k\in[K]}$ that spans the same $K$-dimensional subspace in $\mathbb{R}^d$ as that spanned by $\{\mathfrak{g}_k\}_{k\in[K]}$.

---

[3] Here, orthogonality is in the sense of standard inner product in Euclidean space.

To this aim, the server exploits a modified Gram–Schmidt orthogonalization process over $\{\mathfrak{g}_k\}_{k\in[K]}$ in the following manner [4]

$$\tilde{\mathfrak{g}_1} = \mathfrak{g}_1/|f_k|^\gamma \tag{5}$$

$$\tilde{\mathfrak{g}_k} = \frac{\mathfrak{g}_k - \sum_{i=1}^{k-1} \operatorname{proj}_{\tilde{\mathfrak{g}_i}}(\mathfrak{g}_k)}{|f_k|^\gamma - \sum_{i=1}^{k-1} \frac{\mathfrak{g}_k \cdot \tilde{\mathfrak{g}_i}}{\tilde{\mathfrak{g}_i} \cdot \tilde{\mathfrak{g}_i}}}, \text{ for } k = 2, 3, \ldots, K, \tag{6}$$

where $\gamma > 0$ is a scalar.

• Why such orthogonalization is possible?

First, note that the orthogonalization approach in *phase* 1 is feasible if we assume that the $K$ gradient vectors $\{\mathfrak{g}_k\}_{k\in[K]}$ are linearly independent. Indeed, this assumption is reasonable considering that (i) the gradient vectors $\{\mathfrak{g}_k\}_{k\in[K]}$ are $K$ vectors in $d$-dimensional space, and $d >> K$ for the current deep neural networks (DNNs)[5]; and (ii) the random nature of the gradient vectors due to the non-iid distributions of the local datasets. The validity of this assumption is further confirmed in our thorough experiments over different datasets and models.

### 4.2.2 *Phase* 2, finding optimal $\lambda^*$

In this phase, we aim to find the minimum-norm vector in the convex hull of the mutually-orthogonal gradients found in *Phase (I)*.

First, denote by $\tilde{\mathcal{G}}$ the convex hull of gradient vectors $\{\tilde{\mathfrak{g}_k}\}_{k\in[K]}$ obtained in *Phase* 1; that is,

$$\tilde{\mathcal{G}} = \{\boldsymbol{g} \in \mathbb{R}^d | \boldsymbol{g} = \sum_{k=1}^K \lambda_k \tilde{\mathfrak{g}_k}; \ \lambda_k \geq 0; \ \sum_{k=1}^K \lambda_k = 1\}. \tag{7}$$

In the following, we find the minimal-norm element in $\tilde{\mathcal{G}}$, and then we show that this element is a descent direction for all the objective functions.

Denote by $\boldsymbol{\lambda}^*$ the weights corresponding to the minimal-norm vector in $\tilde{\mathcal{G}}$. To find the weight vector $\boldsymbol{\lambda}^*$, we solve

$$\boldsymbol{g}^* = \arg \min_{\boldsymbol{g} \in \mathcal{G}} \|\boldsymbol{g}\|_2^2, \tag{8}$$

which accordingly finds $\boldsymbol{\lambda}^*$. For an element $\boldsymbol{g} \in \mathcal{G}$, we have

$$\|\boldsymbol{g}\|_2^2 = \|\sum_{k=1}^K \lambda_k \tilde{\mathfrak{g}_k}\|_2^2 = \sum_{k=1}^K \lambda_k^2 \|\tilde{\mathfrak{g}_k}\|_2^2, \tag{9}$$

where we used the fact that $\{\tilde{\mathfrak{g}_k}\}_{k\in[K]}$ are orthogonal.

To solve Equation (8), we first ignore the inequality $\lambda_k \geq 0$, for $k \in [K]$, and then we observe that this constraint will be automatically satisfied. Therefore, we make the following Lagrangian to solve the minimization problem in Equation (8):

$$L(\tilde{\mathfrak{g}}, \boldsymbol{\lambda}) = \|\boldsymbol{g}\|_2^2 - \alpha \left( \sum_{k=1}^K \lambda_k - 1 \right) = \sum_{k=1}^K \lambda_k^2 \|\tilde{\mathfrak{g}_k}\|_2^2 - \alpha \left( \sum_{k=1}^K \lambda_k - 1 \right). \tag{10}$$

Hence,

$$\frac{\partial L}{\partial \lambda_k} = 2\lambda_k \|\tilde{\mathfrak{g}_k}\|_2^2 - \alpha, \tag{11}$$

---

[4]The reason for such normalization is to satisfy *Conditions I* and *II*. This will be proven later in this section.

[5]Also, note that to tackle non-iid distribution of user-specific data, it is a common practice that server selects a different subset of clients in each round (McMahan et al., 2017).

and by setting Equation (11) to zero, we obtain:

$$\lambda_k^* = \frac{\alpha}{2\|\tilde{\mathfrak{g}}_k\|_2^2}. \tag{12}$$

On the other hand, since $\sum_{k=1}^K \lambda_k = 1$, from Equation (12) we obtain

$$\alpha = \frac{2}{\sum_{k=1}^K \frac{1}{\|\tilde{\mathfrak{g}}_k\|_2^2}}, \tag{13}$$

from which the optimal $\boldsymbol{\lambda}^*$ is obtained as follows

$$\lambda_k^* = \frac{1}{\|\tilde{\mathfrak{g}}_k\|_2^2 \sum_{k=1}^K \frac{1}{\|\tilde{\mathfrak{g}}_k\|_2^2}}, \quad \text{for } k \in [K]. \tag{14}$$

Note that $\lambda_k^* > 0$, and therefore the minimum norm vector we found belongs to $\mathcal{G}$.

Using the $\boldsymbol{\lambda}^*$ found in (14), we can calculate $\boldsymbol{\mathfrak{d}}_t = \sum_{k=1}^K \lambda_k^* \tilde{\mathfrak{g}}_k$ as the minimum norm element in the convex hull $\tilde{\mathcal{G}}$. In the following (Theorem 4.1), we show that the negate of $\boldsymbol{\mathfrak{d}}_t$ satisfies both *Conditions (I)* and *(II)*.

**Theorem 4.1.** The negate of $\boldsymbol{\mathfrak{d}}_t = \sum_{k=1}^K \lambda_k^* \tilde{\mathfrak{g}}_k$ satisfies both *Conditions (I)* and *(II)*.

*Proof.* We find the directional derivatives of loss functions $\{f_k\}_{k\in[K]}$ over $\boldsymbol{\mathfrak{d}}_t$. For $\forall k \in [K]$ we have

$$\mathfrak{g}_k \cdot \boldsymbol{\mathfrak{d}}_t = \left( \tilde{\mathfrak{g}}_k (|f_k|^\gamma - \sum_{i=1}^{k-1} \frac{\mathfrak{g}_k \cdot \tilde{\mathfrak{g}}_i}{\tilde{\mathfrak{g}}_i \cdot \tilde{\mathfrak{g}}_i}) + \sum_{i=1}^{k-1} \text{proj}_{\tilde{\mathfrak{g}}_i}(\mathfrak{g}_k) \right) \cdot \left( \sum_{i=1}^K \lambda_i^* \tilde{\mathfrak{g}}_i \right) \tag{15}$$

$$= \lambda_k^* \|\tilde{\mathfrak{g}}_k\|_2^2 \left( |f_k|^\gamma - \sum_{i=1}^{k-1} \frac{\mathfrak{g}_k \cdot \tilde{\mathfrak{g}}_i}{\tilde{\mathfrak{g}}_i \cdot \tilde{\mathfrak{g}}_i} \right) + \sum_{i=1}^{k-1} \frac{\mathfrak{g}_k \cdot \tilde{\mathfrak{g}}_i}{\tilde{\mathfrak{g}}_i \cdot \tilde{\mathfrak{g}}_i} \lambda_i^* \|\tilde{\mathfrak{g}}_i\|_2^2 \tag{16}$$

$$= \frac{\alpha}{2} \left( |f_k|^\gamma - \sum_{i=1}^{k-1} \frac{\mathfrak{g}_k \cdot \tilde{\mathfrak{g}}_i}{\tilde{\mathfrak{g}}_i \cdot \tilde{\mathfrak{g}}_i} \right) + \frac{\alpha}{2} \sum_{i=1}^{k-1} \frac{\mathfrak{g}_k \cdot \tilde{\mathfrak{g}}_i}{\tilde{\mathfrak{g}}_i \cdot \tilde{\mathfrak{g}}_i} \tag{17}$$

$$= \frac{\alpha}{2} |f_k|^\gamma = \frac{|f_k|^\gamma}{\sum_{k=1}^K \frac{1}{\|\tilde{\mathfrak{g}}_k\|_2^2}} > 0, \tag{18}$$

where (i) Equation (15) is obtained by using definition of $\tilde{\mathfrak{g}}_k$ in Equation (6), (ii) Equation (16) follows from the orthogonality of $\{\tilde{\mathfrak{g}}_k\}_{k=1}^K$ vectors, and (iii) Equation (17) is obtained by using Equation (12).

As seen in Equation (18), the directional derivatives over $\boldsymbol{\mathfrak{d}}_t$ are positive, meaning that the direction of $-\boldsymbol{\mathfrak{d}}_t$ is descent for all $\{f_k\}_{k\in[K]}$. In addition, the value of these directional derivatives are proportional to $|f_k|^\gamma$. This implies that if the server changes the global model in the direction of $\boldsymbol{\mathfrak{d}}_t$, the rate of decrease is higher for those functions with larger loss function values. Thus, $-\boldsymbol{\mathfrak{d}}_t$ satisfies both *Conditions (I)* and *(II)*. $\qquad\square$

**Remark 4.2.** As seen, Equation (14) yields a closed-form formula to find the optimal weights for the orthogonal scaled gradients $\{\tilde{\mathfrak{g}}_k\}_{k\in[K]}$, based on which the common direction is obtained. On the contrary, FedMGDA+ (Hu et al., 2022) solves an iterative algorithm to find the updating directions. The complexity of such algorithms is greatly controlled by the size of the model (and the number of participating devices). As the recent DNNs are large in size, deploying such iterative algorithms slows down the FL process. Furthermore, we note that the computational cost of proposed algorithm is negligible (see Appendix F for details).

## 5 The AdaFed algorithm

At iteration $t$, the server computes $\boldsymbol{\mathfrak{d}}_t$ using the methodology described in Section 4.2, and then updates the global model as

$$\boldsymbol{\theta}_{t+1} = \boldsymbol{\theta}_t - \eta_t \boldsymbol{\mathfrak{d}}_t. \tag{19}$$

Similarly to the conventional GD, we note that updating the global model as (19) is a *necessary* condition to have $\boldsymbol{f}(\boldsymbol{\theta}_{t+1}) \leq \boldsymbol{f}(\boldsymbol{\theta}_t)$. In Theorem 5.1, we state the *sufficient* condition to satisfy $\boldsymbol{f}(\boldsymbol{\theta}_{t+1}) \leq \boldsymbol{f}(\boldsymbol{\theta}_t)$.

**Theorem 5.1.** Assume that $\boldsymbol{f} = \{f_k\}_{k\in[K]}$ are L-Lipschitz smooth. If the step-size $\eta_t \in [0, \frac{2}{L}\min\{|f_k|^\gamma\}_{k\in[K]}]$, then $\boldsymbol{f}(\boldsymbol{\theta}_{t+1}) \le \boldsymbol{f}(\boldsymbol{\theta}_t)$, and equality is achieved iff $\boldsymbol{\mathfrak{d}}_t = \boldsymbol{0}$.

*Proof.* If all the $\{f_k\}_{k\in[K]}$ are $L$-smooth, then

$$\boldsymbol{f}(\boldsymbol{\theta}_{t+1}) \le \boldsymbol{f}(\boldsymbol{\theta}_t) + \mathfrak{g}^T(\boldsymbol{\theta}_{t+1} - \boldsymbol{\theta}_t) + \frac{L}{2}\|\boldsymbol{\theta}_{t+1} - \boldsymbol{\theta}_t\|_2^2. \tag{20}$$

Now, for client $k \in [K]$, by using the update rule Equation (19) in Equation (20) we obtain

$$f_k(\boldsymbol{\theta}_{t+1}) \le f_k(\boldsymbol{\theta}_t) - \eta_t \mathfrak{g}_k \cdot \boldsymbol{\mathfrak{d}}_t + \eta_t^2 \frac{L}{2}\|\boldsymbol{\mathfrak{d}}_t\|_2^2. \tag{21}$$

To impose $f_k(\boldsymbol{\theta}_{t+1}) \le f_k(\boldsymbol{\theta}_t)$, we should have

$$\eta_t \mathfrak{g}_k \cdot \boldsymbol{\mathfrak{d}}_t \ge \eta_t^2 \frac{L}{2}\|\boldsymbol{\mathfrak{d}}_t\|_2^2 \quad \Leftrightarrow \quad \mathfrak{g}_k \cdot \boldsymbol{\mathfrak{d}}_t \ge \frac{\eta_t L}{2}\sum_{k=1}^K \frac{\|\tilde{\mathfrak{g}}_k\|_2^2}{\|\tilde{\mathfrak{g}}_k\|_2^4 \left(\sum_{i=1}^K \frac{1}{\|\tilde{\mathfrak{g}}_i\|_2^2}\right)^2} \tag{22}$$

$$\Leftrightarrow \quad \frac{|f_k|^\gamma}{\sum_{k=1}^K \frac{1}{\|\tilde{\mathfrak{g}}_k\|_2^2}} \ge \frac{\eta_t L}{2} \frac{1}{\left(\sum_{k=1}^K \frac{1}{\|\tilde{\mathfrak{g}}_k\|_2^2}\right)^2} \sum_{k=1}^K \frac{1}{\|\tilde{\mathfrak{g}}_k\|_2^2} \tag{23}$$

$$\Leftrightarrow \quad \eta_t \le \frac{2}{L}|f_k|^\gamma. \tag{24}$$

Therefore, if the step-size $\eta_t \in [0, \frac{2}{L}\min\{|f_k|^\gamma\}_{k\in[K]}]$, then $\boldsymbol{f}(\boldsymbol{\theta}_{t+1}) \le \boldsymbol{f}(\boldsymbol{\theta}_t)$. $\qquad\square$

Lastly, similar to many recent FL algorithms McMahan et al. (2017); Li et al. (2019a); Hu et al. (2022), we allow each client to perform a couple of local epochs $e$ before sending its gradient update to the server. In this case, the pseudo-gradients (the opposite of the local updates) will be abusively used as the gradient vectors. It is important to note that we provide a convergence guarantee for this scenario in Section 5.1. We summarize AdaFed in Algorithm 1.

**Remark 5.2.** When $e > 1$, an alternative approach is to use the accumulated loss rather than the loss from the last iteration in line (9) of Algorithm 1. However, based on our experiments, we observed that using the accumulated loss does not affect the overall performance of the algorithm, including its convergence speed, accuracy and fairness. This stands in contrast to the use of pseudo-gradients, which serves clear purposes of accelerating convergence and reducing communication costs.

## 5.1 Convergence results

In the following, we prove the convergence guarantee of AdaFed based on how the clients update the local models: (i) using SGD with $e = 1$, (ii) using GD with $e > 1$, and (iii) using GD with $e = 1$. Of course the strongest convergence guarantee is provided for the latter case.

**Theorem 5.3** ($e = 1$ & local SGD)**.** Assume that $\boldsymbol{f} = \{f_k\}_{k\in[K]}$ are l-Lipschitz continuous and L-Lipschitz smooth, and that the global step-size $\eta_t$ satisfies the following three conditions: (i) $\eta_t \in (0, \frac{1}{2L}]$, (ii) $\lim_{T\to\infty}\sum_{t=0}^T \eta_t \to \infty$, and (iii) $\lim_{T\to\infty}\sum_{t=0}^T \eta_t\sigma_t < \infty$; where $\sigma_t^2 = \mathbf{E}[\|\tilde{\mathfrak{g}}\boldsymbol{\lambda}^* - \tilde{\mathfrak{g}}_s\boldsymbol{\lambda}_s^*\|]^2$ is the variance of stochastic common descent direction. Then

$$\lim_{T\to\infty} \min_{t=0,\dots,T} \mathbf{E}[\|\boldsymbol{\mathfrak{d}}_t\|] \to 0. \tag{25}$$

**Theorem 5.4** ($e > 1$ & local GD)**.** Assume that $\boldsymbol{f} = \{f_k\}_{k\in[K]}$ are l-Lipschitz continuous and L-Lipschitz smooth. Denote by $\eta_t$ and $\eta$ the global and local learning rates, respectively. Also, define $\zeta_t = \|\boldsymbol{\lambda}^* - \boldsymbol{\lambda}_e^*\|$, where $\boldsymbol{\lambda}_e^*$ is the optimum weights obtained from pseudo-gradients after $e$ local epochs. We have

$$\lim_{T\to\infty} \min_{t=0,\dots,T} \|\boldsymbol{\mathfrak{d}}_t\| \to 0, \tag{26}$$

---

**Algorithm 1** AdaFed

---

1: **Input:** Number of global epochs $T$, number of local epochs $e$, global learning rate $\eta_t$, local learning rate $\eta$, initial global model $\boldsymbol{\theta}_0$, local datasets $\{\mathcal{D}_k\}_{k \in K}$.
2: **for** $t = 0, 1, \ldots, T-1$ **do**
3:     Server randomly selects a subset of devices $\mathcal{S}_t$ and sends $\boldsymbol{\theta}_t$ to them.
4:     **for** device $k \in \mathcal{S}_t$ in parallel **do** [local training]
5:         Store the value $\boldsymbol{\theta}_t$ in $\boldsymbol{\theta}_{\text{init}}$; that is $\boldsymbol{\theta}_{\text{init}} \leftarrow \boldsymbol{\theta}_t$.
6:         **for** $e$ epochs **do**
7:             Perform (stochastic) gradient descent over local dataset $\mathcal{D}_k$ to update: $\boldsymbol{\theta}_t \leftarrow \boldsymbol{\theta}_t - \eta \nabla f_k(\boldsymbol{\theta}_t, \mathcal{D}_k)$.
8:         **end for**
9:         Send the pseudo-gradient $\mathfrak{g}_k := \boldsymbol{\theta}_{\text{init}} - \boldsymbol{\theta}_t$ and local loss value $f_k(\boldsymbol{\theta}_t)$ to the server.
10:     **end for**
11:     **for** $k = 1, 2, \ldots, |\mathcal{S}_t|$ **do**
12:         Find $\tilde{\mathfrak{g}}_k$ form Equations (5) and (6).
13:     **end for**
14:     Find $\boldsymbol{\lambda}^*$ from Equation (14).
15:     Calculate $\boldsymbol{\mathfrak{d}}_t := \sum_{k=1}^{K} \lambda_k^* \tilde{\mathfrak{g}}_k$.
16:     $\boldsymbol{\theta}_{t+1} \leftarrow \boldsymbol{\theta}_t - \eta_t \boldsymbol{\mathfrak{d}}_t$.
17: **end for**
18: **Output:** Global model $\boldsymbol{\theta}_T$.

---

if the following conditions are satisfied: (i) $\eta_t \in (0, \frac{1}{2L}]$, (ii) $\lim_{T \to \infty} \sum_{t=0}^{T} \eta_t \to \infty$, (iii) $\lim_{t \to \infty} \eta_t \to 0$, (iv) $\lim_{t \to \infty} \eta \to 0$, and (v) $\lim_{t \to \infty} \zeta_t \to 0$.

Before introducing Theorem 5.5, we first introduce some notations. Denote by $\vartheta$ the Pareto-stationary solution set[6] of minimization problem $\arg\min_{\boldsymbol{\theta}} \boldsymbol{f}(\boldsymbol{\theta})$. Then, denote by $\boldsymbol{\theta}_t^*$ the projection of $\boldsymbol{\theta}_t$ onto the set $\vartheta$; that is, $\boldsymbol{\theta}_t^* = \arg\min_{\boldsymbol{\theta} \in \vartheta} \|\boldsymbol{\theta}_t - \boldsymbol{\theta}\|_2^2$.

**Theorem 5.5** ($e = 1$ & local GD). Assume that $\boldsymbol{f} = \{f_k\}_{k \in [K]}$ are l-Lipschitz continuous and $\sigma$-convex, and that the global step-size $\eta_t$ satisfies the following two conditions: (i) $\lim_{t \to \infty} \sum_{j=0}^{t} \eta_j \to \infty$, and (ii) $\lim_{t \to \infty} \sum_{j=0}^{t} \eta_j^2 < \infty$. Then almost surely $\boldsymbol{\theta}_t \to \boldsymbol{\theta}_t^*$; that is,

$$\mathbb{P}\left(\lim_{t \to \infty} (\boldsymbol{\theta}_t - \boldsymbol{\theta}_t^*) = 0\right) = 1, \tag{27}$$

where $\mathbb{P}(E)$ denotes the probability of event $E$.

The proofs for Theorems 5.3 to 5.5 are provided in Appendices A.1 to A.3, respectively, and we further discuss that the assumptions we made in the theorems are common in the FL literature.

Note that all the Theorems 5.3 to 5.5 provide some types of convergence to a Pareto-optimal solution of optimization problem in Equation (1). Specifically, diminishing $\boldsymbol{\mathfrak{d}}_t$ in Theorems 5.3 and 5.4 implies that we are reaching to a Pareto-optimal point (Désidéri, 2009). On the other hand, Theorem 5.5 explicitly provides this convergence guarantee in an almost surely fashion.

## 6 AdaFed features and a comparative analysis with FedAdam

### 6.1 AdaFed features

Aside from satisfying fairness among the users, we mention some notable features of AdaFed in this section.

The inequality $\boldsymbol{f}(\boldsymbol{\theta}_{t+1}) \leq \boldsymbol{f}(\boldsymbol{\theta}_t)$ gives motivation to the clients to participate in the FL task as their loss functions would decrease upon participating. In addition, since the common direction is more inclined toward

---

[6]In general, the Pareto-stationary solution of multi-objective minimization problem forms a set with cardinality of infinity (Mukai, 1980).

that of the gradients of loss functions with bigger values, a new client could possibly join the FL task in the middle of FL process. In this case, for some consecutive rounds, the loss function for the newly-joined client decreases more compared to those for the other clients.

The parameter $\gamma$ is a hyper-parameter of AdaFed. In fact, different $\gamma$ values yield variable levels of fairness. Thus, $\gamma$ should be tuned to achieve the desired fairness level[7]. In general, a moderate $\gamma$ value enforces a larger respective directional derivative for the devices with the worst performance (larger loss functions), imposing more uniformity to the training accuracy distribution.

Lastly, we note that AdaFed is orthogonal to the popular FL methods such as Fedprox (Li et al., 2020b) and q-FFL (Li et al., 2019a). Therefore, it could be combined with the existing FL algorithms to achieve a better performance, especially with those using personalization (Li et al., 2021).

### 6.2 Comparison with FedAdam and its variants

Similarly to AdaFed, there are some FL algorithms in the literature in which the server adaptively updates the global model. Despite this similarity, we note that the purpose of AdaFed is rather different from these algorithms. For instance, let us consider FedAdam (Reddi et al., 2020); this algorithm changes the global update rule of FedAvg from one-step SGD to one-step adaptive gradient optimization by adopting an Adam optimizer on the server side. Specifically, after gathering local pseudo-gradients and finding their average as $\mathfrak{g}_t = \frac{1}{|\mathcal{S}_t|} \sum_{k \in \mathcal{S}_t} \mathfrak{g}_t^k$, the server updates the global model by Adam optimizer:

$$\boldsymbol{m}_t = \beta_1 \boldsymbol{m}_{t-1} + (1 - \beta_1)\mathfrak{g}_t, \tag{28}$$

$$\boldsymbol{v}_t = \beta_2 \boldsymbol{v}_{t-1} + (1 - \beta_2)\mathfrak{g}_t^2, \tag{29}$$

$$\boldsymbol{\theta}_{t+1} = \boldsymbol{\theta}_t + \eta \frac{\boldsymbol{m}_t}{\sqrt{\boldsymbol{v}_t} + \boldsymbol{\epsilon}}, \tag{30}$$

where $\beta_1$ and $\beta_2$ are two hyper-parameters of the algorithm, and $\boldsymbol{\epsilon}$ is used for numerical stabilization purpose. We note that other variants of this algorithm, such as FedAdagrad and FedYogi (Reddi et al., 2020) and FedAMSGrad (Tong et al., 2020), involve slight changes in the variance term $\boldsymbol{v}_t$.

The primary objective of FedAdam (as well as its variants) is to enhance convergence behavior (Reddi et al., 2020); this is achieved by retaining information from previous epochs, which helps prevent significant fluctuations in the server update. In contrast, AdaFed is designed to promote fairness among clients. Such differences could be confirmed by comparing Algorithm 1 with Equations (28) to (30).

## 7 Experiments

In this section, we conclude the paper with several experiments to demonstrate the performance of AdaFed, and compare its effectiveness with state-of-the-art alternatives under some performance metrics.

• **Datasets:** We conduct a thorough set of experiments over **seven** datasets. The results for four datasets, namely CIFAR-10 and CIFAR-100 (Krizhevsky et al., 2009), FEMNIST (Caldas et al., 2018) and Shakespear (McMahan et al., 2017) are reported in this section; and those for Fashion MNIST (Xiao et al., 2017), TinyImageNet (Le & Yang, 2015), CINIC-10 (Darlow et al., 2018) are reported in Appendix D. Particularly, in order to demonstrate the effectiveness of AdaFed in different FL scenarios, for each of the datasets reported in this section, we consider two different FL setups. In addition, we tested the effectiveness of AdaFed over a real-world noisy dataset, namely Clothing1M (Xiao et al., 2015), in Appendix I.

• **Benchmarks:** We compare the performance of AdaFed against various fair FL algorithms in the literature including: q-FFL (Li et al., 2019a), TERM (Li et al., 2020a), FedMGDA+ (Hu et al., 2022), AFL (Mohri et al., 2019), Ditto (Li et al., 2021), FedFA (Huang et al., 2020b), and lastly FedAvg (McMahan et al., 2017). In our experiments, we conduct a grid-search to find the best hyper-parameters for each of the benchmark

---

[7]Most (if not all) of the fair FL methods introduce an extra hyper-parameter to tune in order to establish a trade-off between fairness and accuracy, for instance: (i) $\epsilon$ in FEDMGDA+ (Hu et al., 2022), (ii) $q$ in Q-FFL & q-FFL (Li et al., 2019a), and (iii) $t$ in TERM (Li et al., 2020a). Similarly, AdaFed introduces a new parameter to make this trade-off.

Table 1: Test accuracy on CIFAR-10. The reported results are averaged over 5 different random seeds.

| Algorithm | Setup 1 | | | | Setup 2 | | | |
|---|---|---|---|---|---|---|---|---|
| | $\bar{a}$ | $\sigma_a$ | Worst 5% | Best 5% | $\bar{a}$ | $\sigma_a$ | Worst 10% | Best 10% |
| FedAvg | 46.85 | 3.54 | 19.84 | 69.28 | 63.55 | 5.44 | 53.40 | 72.24 |
| q-FFL | 46.30 | 3.27 | 23.39 | 68.02 | 57.27 | 5.60 | 47.29 | 66.92 |
| FedMGDA+ | 45.34 | 3.37 | 24.00 | 68.51 | 62.05 | 4.88 | 52.69 | 70.77 |
| FedFA | 46.40 | 3.61 | 19.33 | 69.30 | 63.05 | 4.95 | 48.69 | 70.88 |
| TERM | **47.11** | 3.66 | 28.21 | **69.51** | 64.15 | 5.90 | 56.21 | 72.20 |
| Ditto | 46.31 | 3.44 | 27.14 | 68.44 | 63.49 | 5.70 | 55.99 | 71.34 |
| AdaFed | 46.42 | **3.01** | **31.12** | 69.41 | **64.80** | **4.50** | **58.24** | **72.45** |

Table 2: Test accuracy on CIFAR-100. The reported results are averaged over 5 different random seeds.

| Algorithm | Setup 1 | | | | Setup 2 | | | |
|---|---|---|---|---|---|---|---|---|
| | $\bar{a}$ | $\sigma_a$ | Worst 10% | Best 10% | $\bar{a}$ | $\sigma_a$ | Worst 10% | Best 10% |
| FedAvg | 30.05 | 4.03 | 25.20 | 40.31 | 20.15 | 6.40 | 11.20 | 33.80 |
| q-FFL | 28.86 | 4.44 | 25.38 | 39.77 | **20.20** | 6.24 | 11.09 | 34.02 |
| FedMGDA+ | 29.12 | 4.17 | 25.67 | 39.71 | 20.15 | 5.41 | 11.12 | 33.92 |
| AFL | 30.28 | 3.68 | 25.33 | 39.45 | 18.92 | 4.90 | 11.29 | 28.60 |
| TERM | 30.34 | 3.51 | 27.03 | 39.35 | 17.88 | 5.98 | 10.09 | 31.68 |
| Ditto | 29.81 | 3.79 | 26.90 | 39.39 | 17.52 | 5.65 | 10.21 | 31.25 |
| AdaFed | **31.42** | **3.03** | **28.91** | **40.41** | 20.02 | **4.45** | **11.81** | **34.11** |

methods including AdaFed. The details are reported in Appendix E, and here we only report the results obtained from the best hyper-parameters.

• **Performance metrics:** Denote by $a_k$ the prediction accuracy on device $k$. We use $\bar{a} = \frac{1}{K} \sum_{k=1}^{K} a_k$ as the average test accuracy of the underlying FL algorithm, and use $\sigma_a = \sqrt{\frac{1}{K} \sum_{k=1}^{K} (a_k - \bar{a})^2}$ as the standard deviation of the accuracy across the clients (similarly to (Li et al., 2019a; 2021)). Furthermore, we report worst 10% (5%) and best 10% (5%) accuracies as a common metric in fair FL algorithms (Li et al., 2020a).

• **Notations:** We use **bold** and underlined numbers to denote the best and second best performance, respectively. We use $e$ and $K$ to represent the number of local epochs and that of clients, respectively.

## 7.1 CIFAR-10

CIFAR-10 dataset (Krizhevsky et al., 2009) contains 40K training and 10K test colour images of size $32 \times 32$, which are labeled for 10 classes. The batch size is equal to 64 for both of the following setups.

• **Setup 1:** Following (Wang et al., 2021b), we sort the dataset based on their classes, and then split them into 200 shards. Each client randomly selects two shards without replacement so that each has the same local dataset size. We use a feedforward neural network with 2 hidden layers. We fix $e = 1$ and $K = 100$. We carry out 2000 rounds of communication, and sample 10% of the clients in each round. We run SGD on local datasets with stepsize $\eta = 0.1$.

• **Setup 2:** We distribute the dataset among the clients deploying Dirichlet allocation (Wang et al., 2020) with $\beta = 0.5$. We use ResNet-18 (He et al., 2016) with Group Normalization (Wu & He, 2018). We perform 100 communication rounds in each of which all clients participate. We set $e = 1$, $K = 10$ and $\eta = 0.01$.

Results for both setups are reported in Table 1. Additionally, we depict the average accuracy over the course of training for setup 1 in Appendix G.1.

Table 3: Test accuracy on FEMNIST. The reported results are averaged over 5 different random seeds.

| Algorithm | FEMNIST-**original** | | | | FEMNIST-**skewed** | | | |
| | $\bar{a}$ | $\sigma_a$ | Angle (°) | KL $(a\|u)$ | $\bar{a}$ | $\sigma_a$ | Angle (°) | KL $(a\|u)$ |
|---|---|---|---|---|---|---|---|---|
| FedAvg | 80.42 | 11.16 | 10.18 | 0.017 | 79.24 | 22.30 | 12.29 | 0.054 |
| q-FFL | 80.91 | 10.62 | 9.71 | 0.016 | 84.65 | 18.56 | 12.01 | 0.038 |
| FedMGDA+ | 81.00 | 10.41 | 10.04 | 0.016 | 85.41 | 17.36 | 11.63 | 0.032 |
| TERM | 81.08 | 10.32 | 9.15 | 0.015 | 84.29 | 13.88 | 11.27 | 0.025 |
| AFL | 82.45 | 9.85 | 9.01 | 0.012 | 85.21 | 14.92 | 11.44 | 0.027 |
| Ditto | **83.77** | 10.13 | 9.34 | 0.014 | **92.51** | 14.32 | 11.45 | 0.022 |
| AdaFed | 82.26 | **6.58** | **8.12** | **0.009** | 92.21 | **7.56** | **9.44** | **0.011** |

## 7.2 CIFAR-100

CIFAR-100 (Krizhevsky et al., 2009) contains the same number of samples as those in CIFAR-10, yet it contains 100 classes instead.

The model for both setups is ResNet-18 (He et al., 2016) with Group Normalization (Wu & He, 2018), where all clients participate in each round. We also set $e = 1$ and $\eta = 0.01$. The batch size is equal to 64. The results are reported in Table 2 for both of the following setups:

- **Setup 1**: We set $K = 10$ and $\beta = 0.5$ for Dirichlet allocation, and use 400 communication rounds.

- **Setup 2**: We set $K = 50$ and $\beta = 0.05$ for Dirichlet allocation, and use 200 communication rounds.

Additionally, we perform the same experiments with more local epochs, specifically $e = 10, 20$ as presented in Appendix H.

## 7.3 FEMNIST

FEMNIST (Federated Extended MNIST) Caldas et al. (2018) is a federated image classification dataset distributed over 3,550 devices by the dataset creators. This dataset has 62 classes containing $28 \times 28$-pixel images of digits (0-9) and English characters (A-Z, a-z) written by different people.

For implementation, we use a CNN model with 2 convolutional layers followed by 2 fully-connected layers. The batch size is 32, and $e = 2$ for both of the following setups:

- **FEMNIST-original:** We use the setting in Li et al. (2021), and randomly sample $K = 500$ devices (from the 3550 ones) and train models using the default data stored in each device.

- **FEMNIST-skewed:** Here $K = 100$. We first sample 10 lower case characters ('a'-'j') from Extended MNIST (EMNIST), and then randomly assign 5 classes to each of the 100 devices.

Similarly to (Li et al., 2019a), we use two new fairness metrics for this dataset: (i) the angle between the accuracy distribution and the all-ones vector **1** denoted by Angle (°), and (ii) the KL divergence between the normalized accuracy $a$ and uniform distribution $u$ denoted by KL $(a\|u)$. Results for both setups are reported in Table 3. In addition, we report the distribution of accuracies across clients in Appendix G.2.

## 7.4 Text Data

We use *The Complete Works of William Shakespeare* (McMahan et al., 2017) as the dataset, and train an RNN whose input is 80-character sequence to predict the next character. In this dataset, there are about 1,129 speaking roles. Naturally, each speaking role in the play is treated as a device. Each device stored several text data and those information will be used to train a RNN on each device. The dataset is available on the LEAF website (Caldas et al., 2018). We use $e = 1$, and let all the devices participate in each round. The results are reported in Table 4 for the following two setups:

Table 4: Test accuracy on Shakespeare. The reported results are averaged over 5 different random seeds.

| Algorithm | Setup 1 | | | | Setup 2 | | | |
|---|---|---|---|---|---|---|---|---|
| | $\bar{a}$ | $\sigma_a$ | Worst 10% | Best 10% | $\bar{a}$ | $\sigma_a$ | Worst 10% | Best 10% |
| FedAvg | 53.21 | 9.25 | 51.01 | 54.41 | 50.48 | 1.24 | 48.20 | 52.10 |
| q-FFL | 53.90 | 7.52 | 51.52 | 54.47 | 50.72 | 1.07 | 48.90 | 52.29 |
| FedMGDA+ | 53.08 | 8.14 | 52.84 | 54.51 | 50.41 | 1.09 | 48.18 | 51.99 |
| AFL | 54.58 | 8.44 | 52.87 | 55.84 | 52.45 | 1.23 | 50.02 | 54.17 |
| TERM | 54.16 | 8.21 | 52.09 | 55.15 | 52.17 | 1.11 | 49.14 | 53.62 |
| Ditto | **60.74** | 8.32 | 53.57 | **55.92** | **53.12** | 1.20 | 50.94 | **55.23** |
| AdaFed | 55.65 | **6.55** | **53.79** | 55.86 | 52.89 | **0.98** | **51.02** | 54.48 |

• **Setup 1**: Following McMahan et al. (2017), we subsample 31 speaking roles, and assign each role to a client ($K = 31$) to complete 500 communication rounds. We use a model with two LSTM layers (Hochreiter & Schmidhuber, 1997) and one densely-connected layer. The initial $\eta = 0.8$ with decay rate of 0.95.

• **Setup 2**: Among the 31 speaking roles, the 20 ones with more than 10000 samples are selected, and assigned to 20 clients ($K = 20$). We use one LSTM followed by a fully-connected layer. $\eta = 2$, and the number of communication rounds is 100.

## 7.5 Analysis of Results

Based on Tables 1 to 4, we can attain some notable insights. Compared to other benchmark models, AdaFed leads to significantly more fair solutions. In addition, the average accuracy is not scarified, yet interestingly, for some cases it is improved. We also note that the performance of AdaFed becomes more superior when the level of non-iidness is high. For instance, by referring to FEMNIST-skewed in Table 3, we observe a considerable superiority of AdaFed. Note that the average accuracy of Ditto over FEMNIST is greater than that of AdaFed. This is comprehensible, since Ditto provides a personalized solution to each device, while AdaFed only returns a global parameter $\boldsymbol{\theta}$.

We also observe a similar trend in three other datasets reported in Appendix D. We further analyse the effect of hyper-parameter $\gamma$ in AdaFed in Appendix E.

### 7.5.1 Percentage of improved clients

We measure the training loss before and after each communication round for all participating clients and report the percentage of clients whose loss function decreased or remained unchanged, as defined below

$$\rho_t = \frac{\sum_{k \in \mathcal{S}_t} \mathbb{I}\{\boldsymbol{f}_k(\boldsymbol{\theta}_{t+1}) \leq \boldsymbol{f}_k(\boldsymbol{\theta}_t)\}}{|\mathcal{S}_t|}, \tag{31}$$

where $\mathcal{S}_t$ is the participating clients in round $t$, and $\mathbb{I}(\cdot)$ is the indicator function. Then, we plot $\rho_t$ versus communication rounds for different fair FL benchmarks, including AdaFed. The curves for CIFAR-10 and CIFAR-100 datasets are reported in Figure 2a and Figure 2b, respectively. As seen, both AdaFed and FedMGDA+ consistently outperform other benchmark methods in that fewer clients' performances get worse after participation. This is a unique feature of these two methods. We further note that after enough number of communication rounds, curves for both AdaFed and FedMGDA+ converge to 100% (with a bit of fluctuation).

### 7.5.2 Rate of decrease in loss function

In this part, we observe the loss function values for two clients over the course of training to verify Theorem 4.1. This theorem asserts that the rate of decrease in the loss function is higher for clients with larger initial loss function values. To this end, we select two clients—one with a low initial loss function and one with a high initial loss function—and depict their respective training loss as a function of communication rounds.

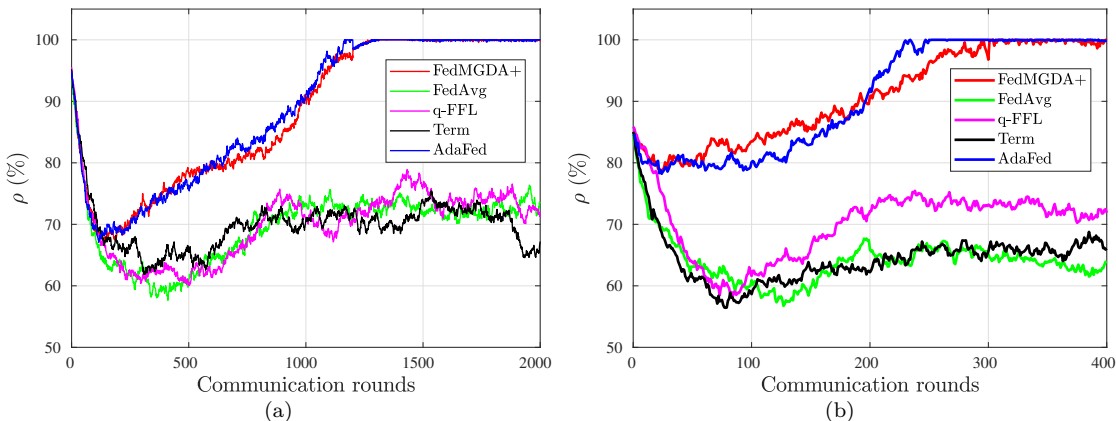

Figure 2: The percentage of improved clients as a function of communication rounds for (a) CIFAR-10 setup one in Section 7.1; and (b) CIFAR-100 setup one in Section 7.2.

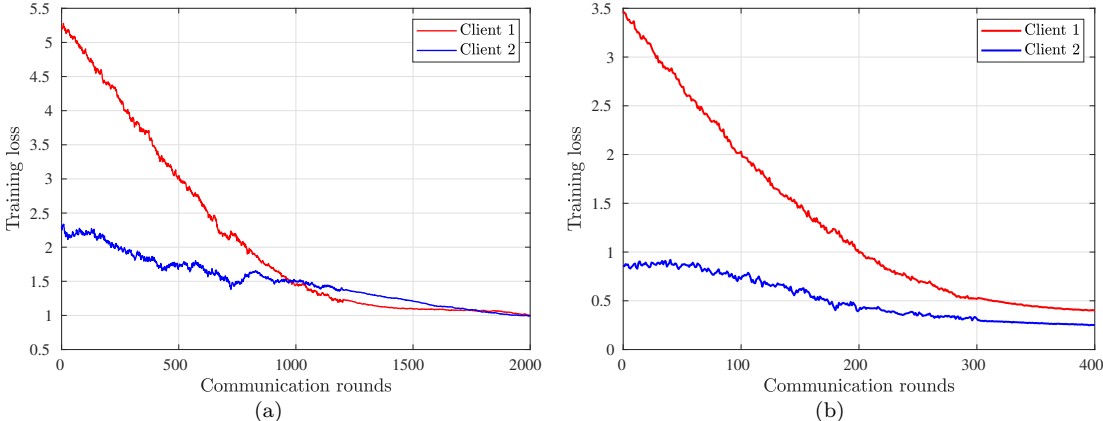

Figure 3: The training loss function for two clients trained in AdaFed framework Vs. the communication rounds for (a) CIFAR-10 setup one in Section 7.1; and (b) CIFAR-100 setup one in Section 7.2.

The curves are illustrated in Figure 3a and Figure 3b for CIFAR-10 and CIFAR-100 datasets, respectively. As observed in both curves, the rate of decrease in the loss function of the client with a larger initial loss function is higher. Additionally, close to the end of the training task, the values for the loss function of the clients converge to almost the same value, indicating fairness among the clients.

### 7.5.3 Convergence of $\|\mathfrak{d}_t\|$

In this part, we aim to observe the behaviour of $\|\mathfrak{d}_t\|$ over the course of training. Particularly, we consider three cases following Theorems 5.3 to 5.5, namely (i) $e = 1$ & local SGD, (ii) $e > 1$ & local GD, and (iii) $e = 1$ & local GD.

For training, we follow the setup in Section 7.1; however, we change $e$ and the local training method—either GD or SGD—to generated the three cases mentioned above. Then, for these three cases, we normalize $\|\mathfrak{d}_t\|$, and depict it versus the communication rounds.

As observed in Figures 4a to 4c, in all the three cases, $\|\mathfrak{d}_t\|$ tends to zero. Nonetheless, the curve for $e = 1$ & local GD is more smooth.

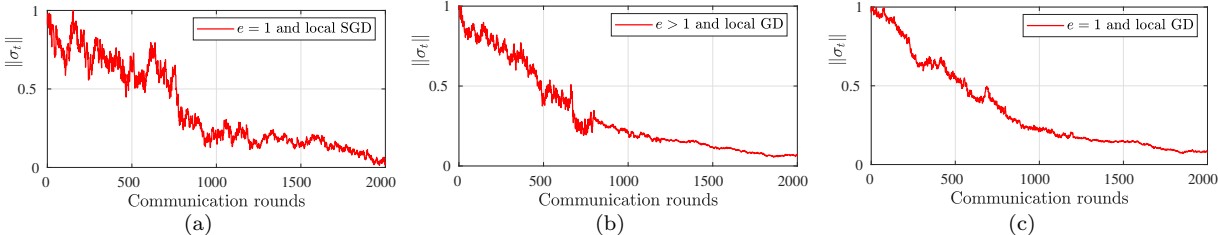

Figure 4: The convergence of $\|\mathfrak{d}_t\|$ as a function of communication rounds for (a) $e = 1$ and local SGD, (b) $e > 1$ and local GD, and (c) $e = 1$ and local GD. The dataset is CIFAR-10.

## 8  Conclusion

In this paper, we proposed a method to enforce fairness in FL task dubbed AdaFed. In AdaFed, the aim is to adaptively tune a common direction along which the server updates the global model. The common direction found by AdaFed enjoys two properties: (i) it is descent for all the local loss functions, and (ii) the loss functions for the clients with worst performance decrease with a higher rate along this direction. These properties were satisfied by using the notion of directional derivative in the multi-objective optimization task. We then derived a closed-form formula for such common direction, and proved that AdaFed converges to a Pareto-stationary point. The effectiveness of AdaFed was demonstrated via thorough experimental results.

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

# A Convergence of AdaFed

In the following, we provide three theorems to analyse the convergence of AdaFed under different scenarios. Specifically, we consider three cases: (i) Theorem A.1 considers $e = 1$ and using SGD for local updates, (ii) Theorem A.2 considers an arbitrary value for $e$ and using GD for local updates, and (iii) Theorem A.4 considers $e = 1$ and using GD for local updates.

## A.1 Case 1: $e = 1$ & local SGD

**Notations:** We use subscript $(\cdot)_s$ to indicate a stochastic value. Using this notation for the values we introduced in the paper, our notations used in the proof of Theorem A.1 are summarized in Table 5.

Table 5: Notations used in Theorem A.1 for $e = 1$ & local SGD.

| Notation | Description |
|:---:|:---|
| $\mathfrak{g}_{k,s}$ | **Stochastic** gradient vector of client $k$. |
| $\mathfrak{g}_s$ | Matrix of **Stochastic** gradient vectors $[\mathfrak{g}_{1,s}, \ldots, \mathfrak{g}_{K,s}]$. |
| $\tilde{\mathfrak{g}}_{k,s}$ | **Stochastic** gradient vector of client $k$ after orthogonalization process. |
| $\tilde{\mathfrak{g}}_s$ | Matrix of orthogonalized **Stochastic** gradient vectors $[\tilde{\mathfrak{g}}_{1,s}, \ldots, \tilde{\mathfrak{g}}_{K,s}]$. |
| $\lambda_{k,s}^*$ | Optimum weights obtained from Equation (14) using **Stochastic** gradients $\tilde{\mathfrak{g}}_s$. |
| $\mathfrak{d}_s$ | Optimum direction obtained using **Stochastic** $\tilde{\mathfrak{g}}_s$; that is, $\mathfrak{d}_s = \sum_{k=1}^{K} \lambda_{k,s}^* \tilde{\mathfrak{g}}_{k,s}$. |

**Theorem A.1.** Assume that $\boldsymbol{f} = \{f_k\}_{k \in [K]}$ are l-Lipschitz continuous and L-Lipschitz smooth, and that the step-size $\eta_t$ satisfies the following three conditions: (i) $\eta_t \in (0, \frac{1}{2L}]$, (ii) $\lim_{T \to \infty} \sum_{t=0}^{T} \eta_t \to \infty$ and (iii) $\lim_{T \to \infty} \sum_{t=0}^{T} \eta_t \sigma_t < \infty$; where $\sigma_t^2 = \mathbf{E}[\|\tilde{\mathfrak{g}}\boldsymbol{\lambda}^* - \tilde{\mathfrak{g}}_s\boldsymbol{\lambda}_s^*\|]^2$ is the variance of stochastic common descent direction. Then

$$\lim_{T \to \infty} \min_{t=0,\ldots,T} \mathbf{E}[\|\mathfrak{d}_t\|] \to 0. \tag{32}$$

*Proof.* Since orthogonal vectors $\{\tilde{\mathfrak{g}}_k\}_{k \in [K]}$ span the same $K$-dimensional space as that spanned by gradient vectors $\{\mathfrak{g}_k\}_{k \in [K]}$, then

$$\exists \{\lambda_k'\}_{k \in [K]} \quad \text{s.t.} \quad \mathfrak{d} = \sum_{k=1}^{K} \lambda_k^* \tilde{\mathfrak{g}}_k = \sum_{k=1}^{K} \lambda_k' \mathfrak{g}_k = \mathfrak{g}\boldsymbol{\lambda}'. \tag{33}$$

Similarly, for the stochastic gradients we have

$$\exists \{\lambda_{k,s}'\}_{k \in [K]} \quad \text{s.t.} \quad \mathfrak{d}_s = \sum_{k=1}^{K} \lambda_{k,s}^* \tilde{\mathfrak{g}}_{k,s} = \sum_{k=1}^{K} \lambda_{k,s}' \mathfrak{g}_{k,s} = \mathfrak{g}_s\boldsymbol{\lambda}_s'. \tag{34}$$

Define $\Delta_t = \mathfrak{g}\boldsymbol{\lambda}' - \mathfrak{g}_s\boldsymbol{\lambda}_s' = \tilde{\mathfrak{g}}\boldsymbol{\lambda}^* - \tilde{\mathfrak{g}}_s\boldsymbol{\lambda}_s^*$, where the last equality is due to the definitions in Equations (33) and (34).

We can find an upper bound for $\boldsymbol{f}(\boldsymbol{\theta}_{t+1})$ as follows

$$\boldsymbol{f}(\boldsymbol{\theta}_{t+1}) = \boldsymbol{f}(\boldsymbol{\theta}_t - \eta_t \mathfrak{d}_t) \tag{35}$$

$$= \boldsymbol{f}(\boldsymbol{\theta}_t - \eta_t \sum_{k=1}^{K} \lambda^*_{k,s} \tilde{\mathfrak{g}}_{k,s}) \tag{36}$$

$$= \boldsymbol{f}(\boldsymbol{\theta}_t - \eta_t \mathbf{g}_s \boldsymbol{\lambda}'_s) \tag{37}$$

$$\leq \boldsymbol{f}(\boldsymbol{\theta}_t) - \eta_t \mathbf{g}^T \mathbf{g}_s^T \boldsymbol{\lambda}'_s + \frac{L\eta_t^2}{2} \|\mathbf{g}_s^T \boldsymbol{\lambda}'_s\|^2 \tag{38}$$

$$\leq \boldsymbol{f}(\boldsymbol{\theta}_t) - \eta_t \mathbf{g}^T \mathbf{g}^T \boldsymbol{\lambda}' + L\eta_t^2 \|\mathbf{g}^T \boldsymbol{\lambda}'\|^2 + \eta_t \mathbf{g}^T \Delta_t + L\eta_t^2 \|\Delta_t\|^2 \tag{39}$$

$$\leq \boldsymbol{f}(\boldsymbol{\theta}_t) - \eta_t(1 - L\eta_t) \|\mathbf{g}^T \boldsymbol{\lambda}'\|^2 + l\eta_t \|\Delta_t\| + L\eta_t^2 \|\Delta_t\|^2, \tag{40}$$

where (36) uses stochastic gradients in the updating rule of AdaFed, (37) is obtained from the definition in (34), (38) holds following the quadratic bound for smooth functions $\boldsymbol{f} = \{f_k\}_{k \in [K]}$, and lastly (40) holds considering the Lipschits continuity of $\boldsymbol{f} = \{f_k\}_{k \in [K]}$.

Assuming $\eta_t \in (0, \frac{1}{2L}]$ and taking expectation from both sides, we obtain:

$$\min_{t=0,\ldots,T} \mathbf{E}[\|\mathfrak{d}_t\|] \leq \frac{\boldsymbol{f}(\boldsymbol{\theta}_0) - \mathbf{E}[\boldsymbol{f}(\boldsymbol{\theta}_{T+1})] + \sum_{t=0}^{T} \eta_t(l\sigma_t + L\eta_t \sigma_t^2)}{\frac{1}{2} \sum_{t=0}^{T} \eta_t}. \tag{41}$$

Using the assumptions (i) $\lim_{T \to \infty} \sum_{j=0}^{T} \eta_t \to \infty$, and (ii) $\lim_{T \to \infty} \sum_{t=0}^{T} \eta_t \sigma_t < \infty$, the theorem will be concluded. Note that *vanishing* $\mathfrak{d}_t$ implies reaching to a Pareto-stationary point of original MoM problem. Yet, the convergence rate is different in different scenarios as we see in the following theorems. □

### A.1.1 Discussing the assumptions

• **The assumptions over the local loss functions:** The two assumptions l-Lipschitz continuous and L-Lipschitz smooth over the local loss functions are two standard assumptions in FL papers providing some sorts of convergence guarantee (Li et al., 2019b).

• **The assumptions over the step-size:** The three assumptions we enforced over the step-size could be easily satisfied as explained in the sequel. For instance, one can pick $\eta_t = \kappa_1 \frac{1}{t}$ for some constant $\kappa_1$ such that $\eta_t \in (0, \frac{1}{2L}]$ is satisfied. Then even if $\sigma_t$ has a extremely loose upper-bound, let's say $\sigma_t < \frac{\kappa_2}{t^\epsilon}$ for a small $\epsilon \in \mathbb{R}_+$ and a constant number $\kappa_2$, then all the three assumptions over the step-size in the theorem will be satisfied. Note that the convergence rate of AdaFed depends on how fast $\sigma_t$ diminishes which depends on how heterogeneous the users are.

### A.2 Case 2: $e > 1$ & local GD

The notations used in this subsection are elaborated in Table 6.

**Theorem A.2.** Assume that $\boldsymbol{f} = \{f_k\}_{k \in [K]}$ are l-Lipschitz continuous and L-Lipschitz smooth. Denote by $\eta_t$ and $\eta$ the global and local learning rate, respectively. Also, define $\zeta_t = \|\boldsymbol{\lambda}^* - \boldsymbol{\lambda}_e^*\|$, where $\boldsymbol{\lambda}_e^*$ is the optimum weights obtained from pseudo-gradients after $e$ local epochs. Then,

$$\lim_{T \to \infty} \min_{t=0,\ldots,T} \|\mathfrak{d}_t\| \to 0, \tag{42}$$

if the following conditions are satisfied: (i) $\eta_t \in (0, \frac{1}{2L}]$, (ii) $\lim_{T \to \infty} \sum_{t=0}^{T} \eta_t \to \infty$ and (iii) $\lim_{t \to \infty} \eta_t \to 0$, (iv) $\lim_{t \to \infty} \eta \to 0$, and (v) $\lim_{t \to \infty} \zeta_t \to 0$.

Table 6: Notations used in the Theorem A.2 for $e > 1$ and local GD.

| Notation | Description |
|---|---|
| $\boldsymbol{\theta}_{(k,e)^t}$ | Updated weight for client $k$ after $e$ local epochs at the $t$-th round of FL. |
| $\mathfrak{g}_{k,e}$ | $\mathfrak{g}_{k,e} = \boldsymbol{\theta}_t - \boldsymbol{\theta}_{(k,e)^t}$; that is, the update vector of client $k$ after $e$ local epochs. |
| $\mathbf{g}_e$ | Matrix of update vectors $[\mathfrak{g}_{1,e}, \ldots, \mathfrak{g}_{K,e}]$. |
| $\tilde{\mathfrak{g}}_{k,e}$ | Update vector of client $k$ after orthogonalization process. |
| $\tilde{\mathbf{g}}_e$ | Matrix of orthogonalized update vectors $[\tilde{\mathfrak{g}}_{1,e}, \ldots, \tilde{\mathfrak{g}}_{K,e}]$. |
| $\lambda_{k,e}^*$ | Optimum weights obtained from Equation (14) using $\tilde{\mathbf{g}}_e$. |
| $\mathfrak{d}_e$ | Optimum direction obtained using $\tilde{\mathbf{g}}_e$; that is, $\mathfrak{d}_e = \sum_{k=1}^{K} \lambda_{k,e}^* \tilde{\mathfrak{g}}_{k,e}$. |

*Proof.* As discussed in the proof of Theorem A.1, we can write

$$\exists \{\lambda_k'\}_{k \in [K]} \quad \text{s.t.} \quad \mathfrak{d} = \sum_{k=1}^{K} \lambda_k^* \tilde{\mathfrak{g}}_k = \sum_{k=1}^{K} \lambda_k' \mathfrak{g}_k = \mathbf{g}\boldsymbol{\lambda}', \tag{43}$$

$$\exists \{\lambda_{k,e}'\}_{k \in [K]} \quad \text{s.t.} \quad \mathfrak{d}_e = \sum_{k=1}^{K} \lambda_{k,e}^* \tilde{\mathfrak{g}}_{k,e} = \sum_{k=1}^{K} \lambda_{k,e}' \mathfrak{g}_{k,e} = \mathbf{g}_e \boldsymbol{\lambda}_e'. \tag{44}$$

To prove Theorem A.2, we first introduce a lemma whose proof is provided in Appendix B.

**Lemma A.3.** Using the notations used in Theorem A.2, and assumming that $\boldsymbol{f} = \{f_k\}_{k \in [K]}$ are L-Lipschitz smooth, we have $\|\mathfrak{g}_{k,e} - \mathfrak{g}_k\| \leq \eta e l$.

Using Lemma A.3, we have

$$\|\mathfrak{d} - \mathfrak{d}_{\boldsymbol{e}}\| = \|\tilde{\mathbf{g}}\boldsymbol{\lambda}^* - \tilde{\mathbf{g}}_e \boldsymbol{\lambda}_e^*\| \leq \|\tilde{\mathbf{g}}\boldsymbol{\lambda}^* - \tilde{\mathbf{g}}\boldsymbol{\lambda}_e^*\| + \|\tilde{\mathbf{g}}\boldsymbol{\lambda}_e^* - \tilde{\mathbf{g}}_e \boldsymbol{\lambda}_e^*\| \tag{45}$$

$$\leq \|\tilde{\mathbf{g}}\|\|\boldsymbol{\lambda}^* - \boldsymbol{\lambda}_e^*\| + \|\mathbf{g}\boldsymbol{\lambda}_e' - \mathbf{g}_e \boldsymbol{\lambda}_e'\| \tag{46}$$

$$\leq \|\tilde{\mathbf{g}}\|\|\boldsymbol{\lambda}^* - \boldsymbol{\lambda}_e^*\| + \eta e l \tag{47}$$

$$\leq \zeta_t l \sqrt{K} + \eta e l, \tag{48}$$

where Equation (45) follows triangular inequality, Equation (46) is obtained from Equations (43) and (44), and Equation (47) uses Lemma A.3.

As seen, if $\lim_{t \to \infty} \eta \to 0$, and $\lim_{t \to \infty} \zeta_t \to 0$, then $\|\mathfrak{d} - \mathfrak{d}_{\boldsymbol{e}}\| \to 0$. Now, by writing the quadratic upper bound we obtain:

$$\boldsymbol{f}(\boldsymbol{\theta}_{t+1}) \leq \boldsymbol{f}(\boldsymbol{\theta}_t) - \eta_t \mathbf{g}^T \mathbf{g}_e^T \boldsymbol{\lambda}_e' + \frac{L\eta_t^2}{2}\|\mathbf{g}_e^T \boldsymbol{\lambda}_e'\|^2 \tag{49}$$

$$\leq \boldsymbol{f}(\boldsymbol{\theta}_t) - \eta_t \mathbf{g}^T \mathbf{g}^T \boldsymbol{\lambda}' + L\eta_t^2 \|\mathbf{g}^T \boldsymbol{\lambda}'\|^2 + \eta_t \mathbf{g}^T (\mathfrak{d} - \mathfrak{d}_{\boldsymbol{e}}) + L\eta_t^2 \|\mathfrak{d} - \mathfrak{d}_{\boldsymbol{e}}\|^2 \tag{50}$$

$$\leq \boldsymbol{f}(\boldsymbol{\theta}_t) - \eta_t(1 - L\eta_t)\|\mathbf{g}^T \boldsymbol{\lambda}'\|^2 + l\eta_t \|\mathfrak{d} - \mathfrak{d}_{\boldsymbol{e}}\| + L\eta_t^2 \|\mathfrak{d} - \mathfrak{d}_{\boldsymbol{e}}\|^2. \tag{51}$$

Noting that $\eta_t \in (0, \frac{1}{2L}]$, and utilizing telescoping yields

$$\min_{t=0,\ldots,T} \|\mathfrak{d}_t\| \leq \frac{\boldsymbol{f}(\boldsymbol{\theta}_0) - \boldsymbol{f}(\boldsymbol{\theta}_{T+1}) + \sum_{t=0}^{T} \eta_t(l\|\mathfrak{d} - \mathfrak{d}_{\boldsymbol{e}}\| + L\eta_t \|\mathfrak{d} - \mathfrak{d}_{\boldsymbol{e}}\|^2)}{\frac{1}{2}\sum_{t=0}^{T} \eta_t}. \tag{52}$$

Using $\|\mathfrak{d} - \mathfrak{d}_{\boldsymbol{e}}\| \to 0$, the Theorem A.2 is concluded. □

### A.3 Case 3: $e = 1$ & local GD

Denote by $\vartheta$ the Pareto-stationary solution set of minimization problem $\arg\min_{\boldsymbol{\theta}} \boldsymbol{f}(\boldsymbol{\theta})$. Then, define $\boldsymbol{\theta}_t^* = \arg\min_{\boldsymbol{\theta} \in \vartheta} \|\boldsymbol{\theta}_t - \boldsymbol{\theta}\|_2^2$.

**Theorem A.4.** Assume that $\boldsymbol{f} = \{f_k\}_{k \in [K]}$ are l-Lipschitz continuous and $\sigma$-convex, and that the step-size $\eta_t$ satisfies the following two conditions: (i) $\lim_{t \to \infty} \sum_{j=0}^t \eta_j \to \infty$ and (ii) $\lim_{t \to \infty} \sum_{j=0}^t \eta_j^2 < \infty$. Then almost surely $\boldsymbol{\theta}_t \to \boldsymbol{\theta}_t^*$; that is,

$$\mathbb{P}\left(\lim_{t \to \infty}(\boldsymbol{\theta}_t - \boldsymbol{\theta}_t^*) = 0\right) = 1, \tag{53}$$

where $\mathbb{P}(E)$ denotes the probability of event $E$.

*Proof.* The proof is inspired from Mercier et al. (2018). Without loss of generality, we assume that all users participate in all rounds.

Based on the definition of $\boldsymbol{\theta}_t^*$ we can say

$$\|\boldsymbol{\theta}_{t+1} - \boldsymbol{\theta}_{t+1}^*\|_2^2 \leq \|\boldsymbol{\theta}_{t+1} - \boldsymbol{\theta}_t^*\|_2^2 = \|\boldsymbol{\theta}_t - \eta_t \mathfrak{d}_t - \boldsymbol{\theta}_t^*\|_2^2 \tag{54}$$

$$= \|\boldsymbol{\theta}_t - \boldsymbol{\theta}_t^*\|_2^2 - 2\eta_t(\boldsymbol{\theta}_t - \boldsymbol{\theta}_t^*) \cdot \mathfrak{d}_t + \eta_t^2 \|\mathfrak{d}_t\|_2^2. \tag{55}$$

To bound the third term in Equation (55), we note that from Equation (23), we have:

$$\eta_t^2 \|\mathfrak{d}_t\|_2^2 = \frac{\eta_t^2}{\sum_{k=1}^K \frac{1}{\|\tilde{\mathfrak{g}}_k\|_2^2}} \leq \frac{\eta_t^2 l^2}{K}. \tag{56}$$

To bound the second term, first note that since orthogonal vectors $\{\tilde{\mathfrak{g}}_k\}_{k \in [K]}$ span the same $K$-dimensional space as that spanned by gradient vectors $\{\mathfrak{g}_k\}_{k \in [K]}$, then

$$\exists \{\lambda_k'\}_{k \in [K]} \text{ s.t. } \mathfrak{d} = \sum_{k=1}^K \lambda_k^* \tilde{\mathfrak{g}}_k = \sum_{k=1}^K \lambda_k' \mathfrak{g}_k. \tag{57}$$

Using Equation (57) and the $\sigma$-convexity of $\{f_k\}_{k \in [K]}$ we obtain

$$(\boldsymbol{\theta}_t - \boldsymbol{\theta}_t^*) \cdot \mathfrak{d}_t = (\boldsymbol{\theta}_t - \boldsymbol{\theta}_t^*) \cdot \sum_{k=1}^K \lambda_k^* \tilde{\mathfrak{g}}_k \tag{58}$$

$$= (\boldsymbol{\theta}_t - \boldsymbol{\theta}_t^*) \cdot \sum_{k=1}^K \lambda_k' \mathfrak{g}_k \tag{59}$$

$$\geq \sum_{k=1}^K \lambda_k' \left(f_k(\boldsymbol{\theta}_t) - f_k(\boldsymbol{\theta}_t^*)\right) + \sigma \frac{\|\boldsymbol{\theta}_t - \boldsymbol{\theta}_t^*\|_2^2}{2} \tag{60}$$

$$\geq \frac{\lambda_\alpha' M}{2} \|\boldsymbol{\theta}_t - \boldsymbol{\theta}_t^*\|_2^2 + \sigma \frac{\|\boldsymbol{\theta}_t - \boldsymbol{\theta}_t^*\|_2^2}{2} \tag{61}$$

$$= \frac{\lambda_\alpha' M + \sigma}{2} \|\boldsymbol{\theta}_t - \boldsymbol{\theta}_t^*\|_2^2. \tag{62}$$

Now, we return back to Equation (55) and find the conditional expectation w.r.t. $\boldsymbol{\theta}_t$ as follows

$$\mathbf{E}[\|\boldsymbol{\theta}_{t+1} - \boldsymbol{\theta}_{t+1}^*\|_2^2 \mid \boldsymbol{\theta}_t] \leq (1 - \eta_t \mathbf{E}[\lambda_\alpha' M + \sigma|\boldsymbol{\theta}_t])\|\boldsymbol{\theta}_t - \boldsymbol{\theta}_t^*\|_2^2 + \frac{\eta_t^2 l^2}{K}. \tag{63}$$

Assume that $\mathbf{E}[\lambda_\alpha' M + \sigma|\boldsymbol{\theta}_t] \geq c$, taking another expectation we obtain:

$$\mathbf{E}[\|\boldsymbol{\theta}_{t+1} - \boldsymbol{\theta}_{t+1}^*\|_2^2] \leq (1 - \eta_t c)\mathbf{E}[\|\boldsymbol{\theta}_t - \boldsymbol{\theta}_t^*\|_2^2] + \frac{\eta_t^2 l^2}{K}, \tag{64}$$

which is a recursive expression. By solving Equation (64) we obtain

$$\mathbf{E}[\|\boldsymbol{\theta}_{t+1} - \boldsymbol{\theta}_{t+1}^*\|_2^2] \leq \underbrace{\prod_{j=0}^{t}(1 - \eta_j c)\mathbf{E}[\|\boldsymbol{\theta}_0 - \boldsymbol{\theta}_0^*\|_2^2]}_{\text{First term}} + \underbrace{\sum_{m=1}^{t} \frac{\prod_{j=1}^{t}(1 - \eta_j c)\eta_m^2 l^2}{K \prod_{j=1}^{m}(1 - \eta_j c)}}_{\text{Second term}}. \tag{65}$$

It is observed that if the limit of both First term and Second term in Equation (65) go to zero, then $\mathbf{E}[\|\boldsymbol{\theta}_{t+1} - \boldsymbol{\theta}_{t+1}^*\|_2^2] \to 0$. For the First term, from the arithmetic-geometric mean inequality we have

$$\lim_{t \to \infty} \prod_{j=0}^{t}(1 - \eta_j c) \leq \lim_{t \to \infty} \left( \frac{\sum_{j=0}^{t}(1 - \eta_j c)}{t} \right)^t = \lim_{t \to \infty} \left( 1 - c\frac{\sum_{j=0}^{t} \eta_j}{t} \right)^t \tag{66}$$

$$= \lim_{t \to \infty} e^{-c\sum_{j=0}^{t} \eta_j}. \tag{67}$$

From Equation (67) it is seen that if $\lim_{t \to \infty} \sum_{j=0}^{t} \eta_j \to \infty$, then the First term is also converges to zero as $t \to \infty$.

On the other hand, consider the Second term in Equation (65). Obviously, if $\lim_{t \to \infty} \sum_{j=0}^{t} \eta_j^2 < \infty$, then the Second term converges to zero as $t \to \infty$.

Hence, if (i) $\lim_{t \to \infty} \sum_{j=0}^{t} \eta_j \to \infty$ and (ii) $\lim_{t \to \infty} \sum_{j=0}^{t} \eta_j^2 < \infty$, then $\mathbf{E}[\|\boldsymbol{\theta}_{t+1} - \boldsymbol{\theta}_{t+1}^*\|_2^2] \to 0$. Consequently, based on standard supermartingale (Mercier et al., 2018), we have

$$\mathbb{P}\left( \lim_{t \to \infty} (\boldsymbol{\theta}_t - \boldsymbol{\theta}_t^*) = 0 \right) = 1. \tag{68}$$

$\square$

# B    Proof of Lemma A.3

*Proof.*

$$\mathfrak{g}_{k,e} = \boldsymbol{\theta}_t - \boldsymbol{\theta}_{(k,e)^t} = (\boldsymbol{\theta}_t - \boldsymbol{\theta}_{(k,1)^t}) + (\boldsymbol{\theta}_{(k,1)^t} - \boldsymbol{\theta}_{(k,2)^t}) + \cdots + (\boldsymbol{\theta}_{(k,e-1)^t} - \boldsymbol{\theta}_{(k,e)^t}) \tag{69}$$

$$= \mathfrak{g}_k(\boldsymbol{\theta}_t) + \eta\mathfrak{g}_{k,1} + \cdots + \eta\mathfrak{g}_{k,e-1}. \tag{70}$$

Hence,

$$\|\mathfrak{g}_{k,e} - \mathfrak{g}_k\| = \|\eta \sum_{j=1}^{e} \mathfrak{g}_{k,j}\| \leq \eta \sum_{j=1}^{e} \|\mathfrak{g}_{k,j}\| \leq \eta el. \tag{71}$$

$\square$

# C    More about fairness in FL

## C.1    Sources of unfairness in federated learning

Unfairness in FL can arise from various sources and is a concern that needs to be addressed in FL systems. Here are some of the key reasons for unfairness in FL:

1. **Non-Representative Data Distribution**: Unfairness can occur when the distribution of data across participating devices or clients is non-representative of the overall population. Some devices may have more or less relevant data, leading to biased model updates.

2. **Data Bias**: If the data collected or used by different clients is inherently biased due to the data collection process, it can lead to unfairness. For example, if certain demographic groups are underrepresented in the training data of some clients, the federated model may not perform well for those groups.

3. **Heterogeneous Data Sources**: Federated learning often involves data from a diverse set of sources, including different device types, locations, or user demographics. Variability in data sources can introduce unfairness as the models may not generalize equally well across all sources.

4. **Varying Data Quality**: Data quality can vary among clients, leading to unfairness. Some clients may have noisy or less reliable data, while others may have high-quality data, affecting the model's performance.

5. **Data Sampling**: The way data is sampled and used for local updates can introduce unfairness. If some clients have imbalanced or non-representative data sampling strategies, it can lead to biased model updates.

6. **Aggregation Bias**: The learned model may exhibit a bias towards devices with larger amounts of data or, if devices are weighted equally, it may favor more commonly occurring devices.

### C.2 Fairness in conventional ML Vs. FL

The concept of fairness is often used to address social biases or performance disparities among different individuals or groups in the machine learning (ML) literature (Barocas et al., 2017). However, in the context of FL, the notion of fairness differs slightly from traditional ML. In FL, fairness primarily pertains to the consistency of performance across various clients. In fact, the difference in the notion of fairness between traditional ML and FL arises from the distinct contexts and challenges of these two settings:

**1. Centralized vs. decentralized data distribution:**

- In traditional ML, data is typically centralized, and fairness is often defined in terms of mitigating biases or disparities within a single, homogeneous dataset. Fairness is evaluated based on how the model treats different individuals or groups within that dataset.

- In FL, data is distributed across multiple decentralized clients or devices. Each client may have its own unique data distribution, and fairness considerations extend to addressing disparities across these clients, ensuring that the federated model provides uniform and equitable performance for all clients.

**2. Client autonomy and data heterogeneity:**

- In FL, clients are autonomous and may have different data sources, labeling processes, and data collection practices. Fairness in this context involves adapting to the heterogeneity and diversity among clients while still achieving equitable outcomes.

- Traditional ML operates under a centralized, unified data schema and is not inherently designed to handle data heterogeneity across sources.

We should note that in certain cases where devices can be naturally clustered into groups with specific attributes, the definition of fairness in FL can be seen as a relaxed version of that in ML, i.e., we optimize for similar but not necessarily identical performance across devices (Li et al., 2019a).

Nevertheless, despite the differences mentioned above, to maintain consistency with the terminology used in the FL literature and the papers we have cited in the main body of this work, we will continue to use the term "fairness" to denote the uniformity of performance across different devices.

## D Additional three datasets

In this section, we evaluate the performance of AdaFed against some benchmarks over some other datasets, namely Fashion MNIST, CINIC-10, and TinyImageNet whose respective results are reported in Appendices D.1 to D.3.

### D.1 Fashion MNIST

Fashion MNIST (Xiao et al., 2017) is an extension of MNIST dataset (LeCun et al., 1998) with images resized to $32 \times 32$ pixels.

We use a fully-connected neural network with 2 hidden layers, and use the same setting as that used in Li et al. (2019a) for our experiments. We set $e = 1$ and use full batchsize, and use $\eta = 0.1$. Then, we conduct 300 rounds of communications. For the benchmarks, we use the same as those we used for CIFAR-10 experiments. The results are reported in Table 7.

By observing the three different classes reported in Table 7, we observe that the fairness level attained in AdaFed is not limited to a dominate class.

Table 7: Test accuracy on Fashion MNIST. The reported results are averaged over 5 different random seeds.

| ALGORITHM | $\bar{a}$ | $\sigma_a$ | SHIRT | PULLOVER | T-SHIRT |
|---|---|---|---|---|---|
| FEDAVG | **80.42** | 3.39 | 64.26 | **87.00** | 89.90 |
| Q-FFL | 78.53 | 2.27 | 71.29 | 81.46 | 82.86 |
| FEDMGDA+ | 79.29 | 2.53 | 72.46 | 79.74 | 85.66 |
| FEDFA | 80.22 | 3.41 | 63.71 | 86.87 | **89.94** |
| ADAFED | 79.14 | **2.12** | **72.49** | 79.81 | 86.99 |

### D.2 CINIC-10

CINIC-10 (Darlow et al., 2018) has 4.5 times as many images as those in CIFAR-10 dataset (270,000 sample images in total). In fact, it is obtained from ImageNet and CIFAR-10 datasets. As a result, this dataset fits FL scenarios since the constituent elements of CINIC-10 are not drawn from the same distribution. Furthermore, we add more non-iidness to the dataset by distributing the data among the clients using Dirichlet allocation with $\beta = 0.5$.

For the model, we use ResNet-18 with group normalization, and set $\eta = 0.01$. There are 200 communication rounds in which all the clients participate with $e = 1$. Also, $K = 50$. Results are reported in Table 8.

Table 8: Test accuracy on CINIC-10. The reported results are averaged over 5 different random seeds.

| ALGORITHM | $\bar{a}$ | $\sigma_a$ | WORST 10% | BEST 10% |
|---|---|---|---|---|
| Q-FFL | **86.57** | 14.91 | 57.70 | 100.00 |
| DITTO | 86.31 | 15.14 | 56.91 | 100.00 |
| AFL | 86.49 | 15.12 | 57.62 | 100.00 |
| TERM | 86.40 | 15.10 | 57.30 | 100.00 |
| ADAFED | 86.34 | **14.85** | **57.88** | 99.99 |

### D.3 TinyImageNet

Tiny-ImageNet (Le & Yang, 2015) is a subset of ImageNet with 100k samples of 200 classes. We distribute the dataset among $K = 20$ clients using Dirichlet allocation with $\beta = 0.05$

We use ResNet-18 with group normalization, and set $\eta = 0.02$. There are 400 communication rounds in which all the clients participate with $e = 1$. The results are reported in Table 9.

Table 9: Test accuracy on TinyImageNet. The reported results are averaged over 5 different random seeds.

| ALGORITHM | $\bar{a}$ | $\sigma_a$ | WORST 10% | BEST 10% |
|---|---|---|---|---|
| Q-FFL | **18.90** | 3.20 | 13.12 | **23.72** |
| AFL | 16.55 | **2.38** | 12.40 | 20.25 |
| TERM | 16.41 | 2.77 | 11.52 | 21.02 |
| FEDMGDA+ | 14.00 | 2.71 | 9.88 | 19.21 |
| ADAFED | 18.05 | 2.35 | **13.24** | 23.08 |

# E   Experiments details, tuning hyper-parameters

For the benchmark methods and also AdaFed, we used grid-search to find the best hyper-parameters for the underlying algorithms. The parameters we tested for each method are as follows:

- **AdaFed:** $\gamma \in \{0, 0.01, 0.1, 1, 5, 10\}$.

- **q-FFL:** $q \in \{0, 0.001, 0.01, 0.1, 1, 2, 5, 10\}$.

- **TERM:** $t \in \{0.1, 0.5, 1, 2, 5\}$.

- **AFL:** $\eta_t \in \{0.01, 0.05, 0.1, 0.5, 1\}$.

- **Ditto:** $\lambda \in \{0.01, 0.05, 0.1, 0.5, 1, 2, 5\}$.

- **FedMGDA+:** $\epsilon \in \{0.01, 0.05, 0.1, 0.5, 1\}$.

- **FedFA:** $(\alpha, \beta) = \{(0.5, 0.5)\}$, $(\gamma_s, \gamma_c) = \{(0.5, 0.9)\}$.

To have a better understanding about how the parameter $\gamma$ in AdaFed affects the performance of the FL task, we report the results for different values of $\gamma$ in AdaFed in this section.

## E.1   CIFAR-10

The best hyper-parameters for the benchmark methods are: $q = 10$ for q-FFL, $\epsilon = 0.5$ for FedMGDA+, and $(\alpha, \beta) = \{(0.5, 0.5)\}$, $(\gamma_s, \gamma_c) = \{(0.5, 0.9)\}$ for FedFA. The detailed results for different $\gamma$ in AdaFed are reported in Table 10. We used $\gamma = 5$ as the best point for Table 1.

Table 10: Tuning $\gamma$ in AdaFed over CIFAR-10. Reported results are averaged over 5 different random seeds.

| ALGORITHM | SETUP 1 | | | | SETUP 2 | | | |
|---|---|---|---|---|---|---|---|---|
| | $\bar{a}$ | $\sigma_a$ | WORST 5% | BEST 5% | $\bar{a}$ | $\sigma_a$ | WORST 10% | BEST 10% |
| ADAFED$_{\gamma=0}$ | 45.44 | 3.43 | 20.18 | 68.04 | 59.88 | 4.89 | 48.12 | 70.62 |
| ADAFED$_{\gamma=0.01}$ | 45.77 | 3.36 | 23.55 | 68.07 | 60.39 | 4.81 | 49.43 | 70.63 |
| ADAFED$_{\gamma=0.1}$ | 46.01 | 3.18 | 27.12 | 68.12 | 60.98 | 4.70 | 50.91 | 70.70 |
| ADAFED$_{\gamma=1}$ | 46.55 | 3.18 | 27.75 | 68.20 | 63.24 | 4.54 | 54.55 | 71.12 |
| ADAFED$_{\gamma=5}$ | 46.42 | 3.01 | 31.12 | 67.73 | 64.80 | 4.50 | 58.24 | 72.45 |
| ADAFED$_{\gamma=10}$ | 46.00 | 2.88 | 35.21 | 67.35 | 63.25 | 4.66 | 51.74 | 71.25 |

## E.2   CIFAR-100

The best hyper-parameters for the benchmark methods are: $q = 0.1$ for q-FFL, $t = 0.5$ in TERM, and $\eta_t = 0.5$ in AFL. In addition, the detailed results for different $\gamma$ in AdaFed are reported in Table 11. We used $\gamma = 1$ as the best point for Table 2.

Table 11: Tuning $\gamma$ in AdaFed over CIFAR-100. Reported results are averaged over 5 different random seeds.

| Algorithm | Setup 1 | | | | Setup 2 | | | |
|---|---|---|---|---|---|---|---|---|
| | $\bar{a}$ | $\sigma_a$ | Worst 10% | Best 10% | $\bar{a}$ | $\sigma_a$ | Worst 10% | Best 10% |
| AdaFed$_{\gamma=0}$ | 29.41 | 4.45 | 24.41 | 39.21 | 17.05 | 6.71 | 10.04 | 27.41 |
| AdaFed$_{\gamma=0.01}$ | 30.12 | 4.05 | 25.23 | 39.41 | 17.77 | 6.09 | 10.43 | 28.42 |
| AdaFed$_{\gamma=0.1}$ | 31.05 | 3.52 | 26.13 | 40.12 | 19.51 | 4.95 | 10.89 | 32.10 |
| AdaFed$_{\gamma=1}$ | 31.42 | 3.03 | 28.91 | 40.41 | 20.02 | 4.45 | 11.81 | 34.11 |
| AdaFed$_{\gamma=5}$ | 31.23 | 2.95 | 28.12 | 40.20 | 19.79 | 4.31 | 11.86 | 33.67 |
| AdaFed$_{\gamma=10}$ | 31.34 | 2.91 | 28.52 | 40.15 | 19.61 | 4.56 | 11.42 | 32.91 |

### E.3 Fashion MNIST

The best hyper-parameters for the benchmark methods are: $\epsilon = 0.5$ for FedMGDA+, $q = 0.1$ for q-FFL, $(\alpha, \beta) = \{(0.5, 0.5)\}$, $(\gamma_s, \gamma_c) = \{(0.5, 0.9)\}$ for FedFA. The detailed results for different $\gamma$ in AdaFed are reported in Table 12. We used $\gamma = 1$ as the best point for Table 7.

Table 12: Tuning $\gamma$ in AdaFed over Fashion MNIST. Reported results are averaged over 5 different seeds.

| ALGORITHM | $\bar{a}$ | $\sigma_a$ | SHIRT | PULLOVER | T-SHIRT |
|---|---|---|---|---|---|
| ADAFED$_{\gamma=0}$ | 78.84 | 2.55 | 71.77 | 78.34 | 84.12 |
| ADAFED$_{\gamma=0.01}$ | 78.88 | 2.41 | 71.73 | 78.66 | 85.62 |
| ADAFED$_{\gamma=0.1}$ | 79.24 | 2.30 | 72.46 | 79.14 | 85.66 |
| ADAFED$_{\gamma=1}$ | 79.14 | 2.12 | 72.33 | 79.81 | 83.99 |
| ADAFED$_{\gamma=5}$ | 79.04 | 2.09 | 71.55 | 78.37 | 85.41 |
| ADAFED$_{\gamma=10}$ | 78.91 | 1.96 | 71.43 | 78.04 | 85.82 |

### E.4 FEMNIST

The best hyper-parameters for the benchmark methods are: $\lambda = 0.1$ for Ditto, $q = 0.1$ for q-FFL, $t = 0.5$ for TERM, $\eta_t = 0.5$ for AFL. Also, the detailed results for different $\gamma$ in AdaFed are reported in Table 3. We used $\gamma = 1$ as the best point for Table 13.

Table 13: Tuning $\gamma$ in AdaFed over FEMNIST. Reported results are averaged over 5 different seeds.

| Algorithm | FEMNIST-**original** | | | | FEMNIST-**skewed** | | | |
|---|---|---|---|---|---|---|---|---|
| | $\bar{a}$ | $\sigma_a$ | Angle (°) | KL $(a\|u)$ | $\bar{a}$ | $\sigma_a$ | Angle (°) | KL $(a\|u)$ |
| AdaFed$_{\gamma=0}$ | 81.32 | 13.59 | 10.85 | 0.019 | 84.39 | 13.54 | 11.32 | 0.024 |
| AdaFed$_{\gamma=0.01}$ | 82.67 | 12.03 | 10.68 | 0.018 | 87.66 | 12.02 | 10.91 | 0.019 |
| AdaFed$_{\gamma=0.1}$ | 81.60 | 8.72 | 9.23 | 0.011 | 88.62 | 10.59 | 10.75 | 0.017 |
| AdaFed$_{\gamma=1}$ | 82.26 | 6.58 | 8.12 | 0.009 | 92.21 | 7.56 | 9.44 | 0.011 |
| AdaFed$_{\gamma=5}$ | 80.10 | 5.16 | 7.29 | 0.007 | 90.12 | 5.82 | 7.31 | 0.009 |
| AdaFed$_{\gamma=10}$ | 80.05 | 3.03 | 6.44 | 0.007 | 84.38 | 4.49 | 6.99 | 0.008 |

### E.5 Shakespeare

The best hyper-parameters for the benchmark methods are: $q = 0.1$ for q-FFL, $\lambda = 0.1$ for Ditto, and $\eta_t = 0.5$ for AFL. Furthermore, the results obtained for different $\gamma$ values in AdaFed are reported in Table 14. We used $\gamma = 0.1$ as the best point for Table 4.

Table 14: Tuning $\gamma$ in AdaFed over Shakespeare. Reported results are averaged over 5 different seeds.

| Algorithm | Setup 1 | | | | Setup 2 | | | |
|---|---|---|---|---|---|---|---|---|
| | $\bar{a}$ | $\sigma_a$ | Worst 10% | Best 10% | $\bar{a}$ | $\sigma_a$ | Worst 10% | Best 10% |
| AdaFed$_{\gamma=0}$ | 48.40 | 13.5 | 44.12 | 51.29 | 48.80 | 1.58 | 46.23 | 51.12 |
| AdaFed$_{\gamma=0.01}$ | 53.55 | 8.01 | 50.96 | 54.46 | 51.67 | 1.10 | 48.71 | 53.16 |
| AdaFed$_{\gamma=0.1}$ | 55.65 | 6.55 | 53.79 | 55.86 | 52.89 | 0.98 | 51.02 | 54.48 |
| AdaFed$_{\gamma=1}$ | 53.91 | 5.10 | 51.94 | 54.06 | 51.44 | 1.06 | 50.88 | 54.52 |
| AdaFed$_{\gamma=5}$ | 54.40 | 4.15 | 52.17 | 54.77 | 51.20 | 1.05 | 50.72 | 54.61 |
| AdaFed$_{\gamma=10}$ | 54.56 | 4.22 | 52.20 | 54.73 | 51.19 | 1.07 | 50.70 | 54.01 |

### E.6 CINIC-10

The best hyper-parameters for the benchmark methods are: $t = 0.5$ for TERM, $q = 0.1$ for q-FFL, $\lambda = 0.1$ for Ditto, and $\eta_t = 0.5$ for AFL. Furthermore, the results obtained for different $\gamma$ values in AdaFed are reported in Table 15. We used $\gamma = 1$ as the best point for Table 8.

Table 15: Tuning $\gamma$ in AdaFed over CINIC-10. Reported results are averaged over 5 different seeds.

| Algorithm | $\bar{a}$ | $\sigma_a$ | Worst 10% | Best 10% |
|---|---|---|---|---|
| AdaFed$_{\gamma=0}$ | 85.17 | 15.71 | 54.67 | 99.92 |
| AdaFed$_{\gamma=0.01}$ | 85.87 | 15.54 | 56.12 | 99.95 |
| AdaFed$_{\gamma=0.1}$ | 86.13 | 15.32 | 57.01 | 99.98 |
| AdaFed$_{\gamma=1}$ | 86.34 | 14.85 | 57.88 | 99.99 |
| AdaFed$_{\gamma=5}$ | 86.03 | 15.01 | 57.72 | 99.98 |
| AdaFed$_{\gamma=10}$ | 85.49 | 15.08 | 57.23 | 99.99 |

### E.7 TinyImageNet

The best hyper-parameters for the benchmark methods are: $\epsilon = 0.05$ for FedMGDA+, $t = 0.5$ for TERM, $q = 0.1$ for q-FFL, and $\eta_t = 0.5$ for AFL. Furthermore, the results obtained for different $\gamma$ values in AdaFed are reported in Table 16. We used $\gamma = 1$ as the best point for Table 9.

### E.8 The effect of parameter $\gamma$

In this section, we reported the results for AdaFed over different datasets when $\gamma$ takes different values. Based on the tables reported in this section, we observe almost a similar trend over all the dataset. As a rule of thumb, a higher (lower) $\gamma$ yields a higher (lower) fairness and slightly lower (higher) accuracy. Nevertheless, the best performance of AdaFed (in terms of establishing an appropriate trade-off between average accuracy and fairness) is achieved for a moderate value of $\gamma$. This is also consistent with the other fairness methods in the literature, where in most cases, the best hyper-parameter is a moderate one.

Table 16: Tuning $\gamma$ in AdaFed over TinyImageNet. Reported results are averaged over 5 different seeds.

| ALGORITHM | $\bar{a}$ | $\sigma_a$ | WORST 10% | BEST 10% |
|---|---|---|---|---|
| ADAFED$_{\gamma=0}$ | 13.25 | 3.14 | 9.82 | 19.24 |
| ADAFED$_{\gamma=0.01}$ | 14.38 | 2.72 | 10.12 | 19.97 |
| ADAFED$_{\gamma=0.1}$ | 16.20 | 2.65 | 11.65 | 21.12 |
| ADAFED$_{\gamma=1}$ | 18.05 | 2.35 | 13.24 | 23.08 |
| ADAFED$_{\gamma=5}$ | 17.76 | 2.31 | 12.44 | 22.84 |
| ADAFED$_{\gamma=10}$ | 17.05 | 2.38 | 12.58 | 23.67 |

# F  Computation cost of Adafed

## F.1  Comparing to FedMGDA+

First, note that AdaFed concept is built upon that of FedMGDA+ (Hu et al., 2022) (and FairWire in Hamidi & Damen (2024)), in that both use Pareto-optimal notion to enforce fairness in FL task. Note that the optimal solutions in MoM usually forms a set (in general of infinite cardinality). As discussed, what distinguishes FedMGDA+ and AdaFed is that to which point of this set these algorithms converge. Particularly, AdaFed converges to more uniform solutions Figure 1a. This is because FedMGDA+ algorithm only satisfies *Condition (I)*, yet in AdaFed, both *Conditions (I)* and *(II)* are held.

Interestingly, the cost of performing AdaFed is less than that of performing FedMGDA+. To elucidate, FedMGDA+ also finds the minimum-norm vector in the convex hull of the gradients' space in order to find a common descent direction. To this end, they used generic quadratic programming which entails iteratively finding the minimum-norm vector in the convex hull of the local gradients. One of the pros of AdaFed is that it finds the common descent direction without performing any iterations over the gradient vectors. Thus, AdaFed not only yields a higher level of fairness compared to FedMGDA+, but also solves its complexity issue.

## F.2  Running time for AdaFed

Assume that the number of clients is $K$, and the dimension of the gradient vectors is $d$. Then, the orthogonalization for $k$-th client, $k \in [K]$, needs $\mathcal{O}(2dk)$ operations (by operations we meant multiplications and additions). Hence, the total number of operations needed for orthogonalization process in equal to $\mathcal{O}(2dK^2)$ (Also note that Gram-Schmidt is the most efficient algorithm for orthogonalization).

In our experimental setup, we realized that the overhead of AdaFed is negligible, resulting almost the same overall running time for FedAvg and AdaFed. To justify this fact, please refer to FedMGDA+ paper where they discussed the overhead of their proposed method; as they claimed, the overhead is negligible yielding the same running time as FedAVG. On the other hand, as explained in Appendix F.1, the complexity of AdaFed is lower than that of FedMGDA+.

# G  Curves and histograms

## G.1  Training curves for CIFAR-10 and CIFAR-100

In this subsection, we depict the average test accuracy over the course of training for CIFAR-10 dataset using setup one (see Section 7.1). In particular, we depict the average test accuracy across all the clients Vs. the number of communication rounds. Additionally, to demonstrate the convergence of the FL algorithms after 2000 communication rounds, we have depicted the training curve for 4000 rounds.

The curve for each method is obtained by using the best hyper-parameter of the respective method (we discussed the details of hyper-parameter tuning in Appendix E). Furthermore, the curves are averaged over five different seeds.

The results are shown in Figure 5 and Figure 6 for CIFAR-10 and CIFAR-100, respectively. Particularly in Figure 5, AdaFed converges faster than the benchmark methods. Specifically, AdaFed reaches average test accuracy of 40% after around 400 communication rounds; however, the benchmark methods reach this accuracy after around 900 rounds. Indeed, this is another advantage of AdaFed in addition to imposing fairness across the clients.

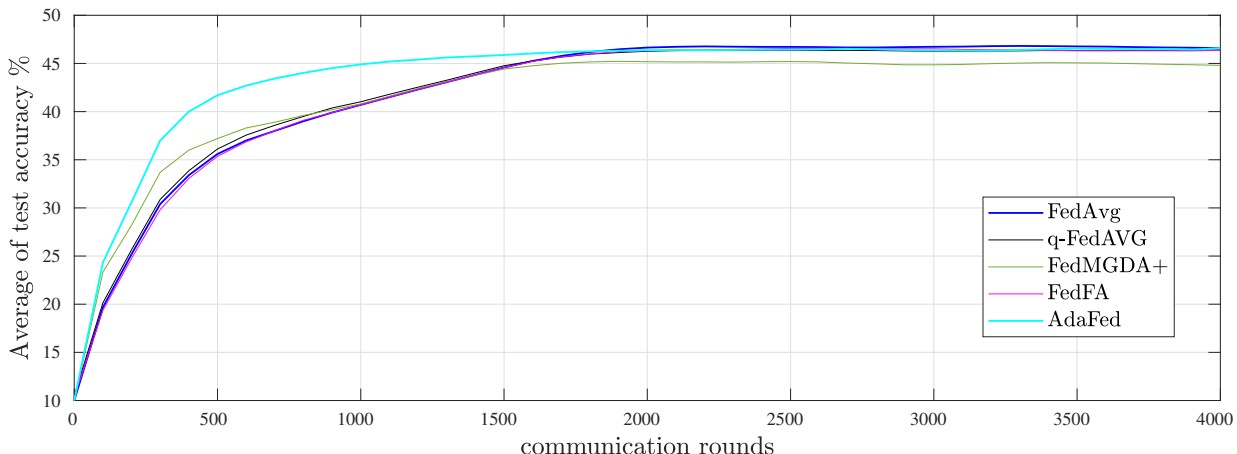

Figure 5: Average test accuracy across clients for different FL methods on CIFAR-10. The setup for the experiments is elaborated in Section 7.1, setup 1.

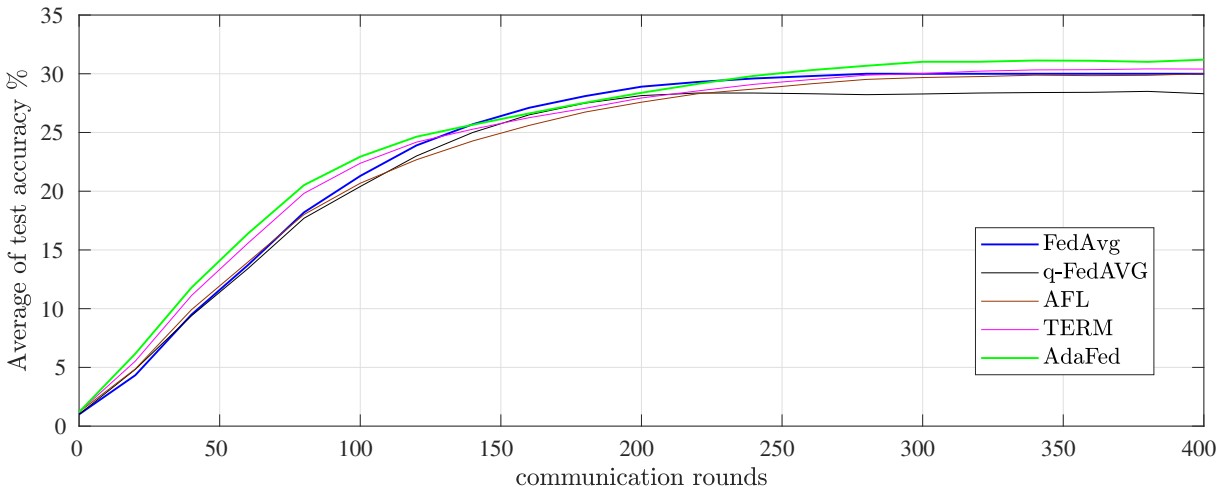

Figure 6: Average test accuracy across clients for different FL methods on CIFAR-100. The setup for the experiments is elaborated in Section 7.2, setup 1.

## G.2  Histogram of accuracies

To better observe the spread of clients accuracy, we depict the histogram of accuracy across 500 clients for the Original FEMNIST dataset (the setup for the experiment is discussed in Appendix D.1). To this end, we depict the histogram of the clients' accuracies for three different methods: (i) FedAvg, (ii) Q-FFL, and (iii)

AdaFed($\gamma = 5$); all using their well-tuned hyper-parameters. The result is depicted in Appendix G.2. As seen, the distribution of the accuracy is more concentrated (fair) for AdaFed.

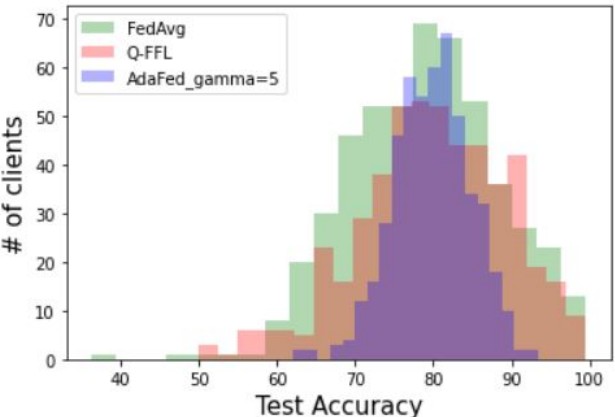

Figure 7: The distribution of clients accuracy for Original FEMNIST dataset using three different methods, namely: (i) FedAvg, (ii) Q-FFL, and (iii) AdaFed.

## H CIFAR-100 results with more local epochs

In this section, we want to test the performance of AdaFed using a larger number of local epochs $e$. To this end, we use the same setups as those used in the main body of the paper to produce the results for CIFAR-100, but we change the number of local epochs $e$ to 10 and 20. [8]. The results for $e = 10$ and $e = 20$ are reported in Table 17 and Table 18, respectively. We highlight two key observations from the tables:

- AdaFed can still provide a higher level of fairness compared to the benchmark methods;

- While increasing the number of local epochs from 1 to 10 results in higher average accuracy, this trend is not observed when further increasing $e$ to 20.

Table 17: Test accuracy on CIFAR-100 with $e = 10$. The reported results are averaged over 5 different random seeds.

| Algorithm | Setup 1 | | | | Setup 2 | | | |
|---|---|---|---|---|---|---|---|---|
| | $\bar{a}$ | $\sigma_a$ | Worst 10% | Best 10% | $\bar{a}$ | $\sigma_a$ | Worst 10% | Best 10% |
| FedAvg | 31.14 | 4.09 | 25.30 | 40.54 | 20.54 | 6.42 | 11.12 | 33.47 |
| q-FFL | 29.45 | 4.66 | 25.35 | 39.91 | **20.77** | 6.20 | 11.05 | 33.52 |
| AFL | 31.17 | 3.69 | 25.12 | 39.52 | 19.32 | 4.85 | 11.23 | 28.93 |
| TERM | 30.56 | 3.63 | 27.19 | 39.46 | 17.91 | 5.87 | 10.11 | 32.00 |
| AdaFed | **31.19** | **3.14** | **28.81** | **40.42** | 20.41 | **4.71** | **11.39** | **34.08** |

## I Integration with a Label Noise Correction method

### I.1 What is label noise in FL?

Label noise in FL refers to inaccuracies or errors in the ground truth labels associated with the data used for training. It occurs when the labels assigned to data points are incorrect or noisy due to various reasons.

---

[8]We selected $e = \{10, 20\}$ because these values have been commonly utilized in the literature for the CIFAR-100 dataset.

Table 18: Test accuracy on CIFAR-100 with $e = 20$. The reported results are averaged over 5 different random seeds.

| Algorithm | Setup 1 | | | | Setup 2 | | | |
|---|---|---|---|---|---|---|---|---|
| | $\bar{a}$ | $\sigma_a$ | Worst 10% | Best 10% | $\bar{a}$ | $\sigma_a$ | Worst 10% | Best 10% |
| FedAvg | 29.11 | 4.31 | 24.61 | 39.45 | 18.05 | 6.12 | 9.15 | 30.12 |
| q-FFL | 29.15 | 4.23 | 25.12 | 39.67 | **19.02** | 6.15 | 8.41 | 31.66 |
| AFL | 30.38 | 3.78 | 25.00 | 39.12 | 17.74 | 4.96 | 10.01 | 27.08 |
| TERM | **31.15** | 3.62 | 27.02 | **40.41** | 15.81 | 5.68 | 8.17 | 29.26 |
| AdaFed | 30.41 | **3.19** | **27.35** | 40.18 | 18.37 | **4.21** | **10.55** | **31.78** |

Label noise can be introduced at different stages of data collection, annotation, or transmission, and it can have a significant impact on the performance and reliability of FL models.

Label noise in FL is particularly challenging to address because FL relies on decentralized data sources, and participants may have limited control over label quality in remote environments. Dealing with label noise often involves developing robust models and FL algorithms that can adapt to the presence of inaccuracies in the labels.

### I.2 Are the fair FL algorithms robust against label noise?

The primary intention of the fair FL algorithms including AdaFed is to ensure fairness among the clients while maintaining the average accuracy across them. Yet, these algorithms are not robust against label noise (mislabeled instances).

Nonetheless, AdaFed could be integrated with label-noise resistant methods in the literature yielding an FL method which (i) satisfies fairness among the clients, and (ii) is robust against the label noise. In particular, among the label-noise resistant FL algorithms in the literature, we select FedCorr (Xu et al., 2022) to be integrated with AdaFed.

FedCorr introduces a dimensionality-based filter to identify noisy clients, which is accomplished by measuring the local intrinsic dimensionality (LID) of local model prediction subspaces. They demonstrate that it is possible to distinguish clean datasets from noisy ones by observing the behavior of LID scores during the training process (we omit further discussions about FedCorr, and refer interested readers to their paper for more details).

Similarly to FedCorr, we use a real-world noisy dataset, namely Clothing1M[9] (Xiao et al., 2015), and we use exactly the same setting as they used for this dataset[10]. In particular, we use local SGD with a momentum of 0.5, with a batch size of 16, and five local epochs, and set the hyper-parameter $T_1 = 2$ in their algorithm. In addition, when integrated with AdaFed, we set $\gamma = 5$ for AdaFed.

The results are summarized in Table 19. As observed, the average accuracy obtained by AdaFed is around 2.2% lower than that obtained from FedCorr which shows that AdaFed is not robust against label-noise. Moreover, as expected AdaFed results in a more fair client accuracy. On the other hand, when AdaFed is combined with FedCorr, the average accuracy improves while maintaining satisfactory fairness among the clients.

---

[9]Clothing1M contains 1M clothing images in 14 classes. It is a dataset with noisy labels, since the data is collected from several online shopping websites and include many mislabelled samples.

[10]https://github.com/Xu-Jingyi/FedCorr

Table 19: Test accuracy on Clothing1M dataset. The reported results are averaged over 5 different random seeds.

| Algorithm | $\bar{a}$ | $\sigma_a$ | Worst 10% | Best 10% |
|---|---|---|---|---|
| FedAvg | 70.49 | 13.25 | 43.09 | 91.05 |
| FedCorr | 72.55 | 13.27 | 43.12 | 91.15 |
| AdaFed | 70.35 | 5.17 | 49.91 | 90.77 |
| FedCorr + AdaFed | 72.29 | 8.12 | 46.52 | 91.02 |

