# OpenReview forum: "AdaFed: Fair Federated Learning via Adaptive Common Descent Direction"
_TMLR — Accepted by TMLR_

### Review · Reviewer_KkcE · 2023-10-01

**Summary Of Contributions:**

This work proposes a way to tackle the fairness problem in federated learning (FL). Defining the problem as the variance in accuracies  across clients, this work employs multi-objective minimization to develop a novel approach of global model aggregation, such that this problem can be addressed. Additionally, some theoretical analysis has been conduct to reason the choice of global learning rate and to prove the convergence. In the experimental section, results of extensive datasets show the promise of the proposed AdaFed approach. In particular, AdaFed performs outstandingly in reducing the variance of accuracies across the clients, i.e. the problem attempt to address in this paper.

**Audience:**

Yes

**Claims And Evidence:**

Yes

**Requested Changes:**

The following changes can strengthen the work in my view:

1. More experiments with a larger number of epochs. Even negative results can add credibility to this work.
2. Discussion regarding the weaknesses.
3. Modern datasets usually contain mis-labeled instances. It seems the proposed approach is susceptible to such problems. Discussion and experiments to show the limitation are preferred.

**Strengths And Weaknesses:**

# Strengths

1. This work successfully correlates the loss variance with the aggregation algorithm using principles derived from multi-objective optimization (MOO). This is a unique and well-crafted concept.
2. Further analysis has been undertaken to rationalize the choice of global learning rate and to prove the convergence, adding to the paper's reliability.
3. This work clearly clarifies how the proposed approach is derived, making the work accessible to the audience not working directly on MOO. This has the advantage of attracting more researchers in the field of FL to study MOO.
4. Experimental results show that the variance in accuracy is efficiently reduced compared with the baselines, proving the effectiveness of AdaFed.

# Weaknesses

1. The derivation of AdaFed is essentially based on distributed SGD instead of FL. The main difference between these two approaches is that one performs aggregation at every iteration, while the other performs aggregation after many epochs of local training. While I believe some insights from distributed SGD can be transferred to FL, they are not always generalizable. This work lacks discussion regarding this.
2. When applied to a real FL application, accumulated gradients are regarded as pseudo-gradients. However, only the loss of the last iteration is used to re-scale the gradient (Line 9, Algorithm 1). Can the author explain the reason of choosing this approach instead of accumulated loss?
3. The experimental results and settings are not very convincing. In particular, the number of epochs is mostly set to 1 which is rare for FL applications due to the concern about communication overhead. Additionally, data splitting does not seem to impose a very difficult learning task and the networks are not small for the respective dataset. However, the reported accuracies are too small.

---

### Review · Reviewer_6yn3 · 2023-10-03

**Summary Of Contributions:**

The authors introduce a new server-side averaging method for federated learning. For each clients' update vector that is sent to the server, a scalar multiplicative value is found such that the average of the scaled update vectors results in a fairer model at convergence.
The authors motivate their method thoroughly, providing analysis and intuition. Their approach is accompanied by convergence proofs (that I have not read in detail), as well as experimental results across a wide variety of datasets & benchmarks.

**Audience:**

Yes

**Broader Impact Concerns:**

No concerns

**Claims And Evidence:**

Yes

**Requested Changes:**

The experimental evaluation is very extensive, but I am surprised by the decision to optimize only 200 or 400 rounds for Cifar100. In my own experience with this dataset and model in the FL context, convergence is achieved only after a higher number of rounds - I have not replicated the exact setting of course, but I would be curious to see a learning curve. Along the same lines Figure 2 for Cifar10 seems to suggest that in terms of average test accuracy, the differently trained algorithms have not converged yet. I would like to understand how much of a compromise AdaFed makes in terms of accuracy compared to these baselines when trained to convergence.

As mentioned above, I would like to understand if adaptive server-side optimizers should not serve as a relevant baseline to these settings and if yes, please include one in your baselines.

That being said, I really enjoyed reading this paper and believe it to be strong.

**Strengths And Weaknesses:**

Strengths:
- Well-motivated, good exposition, good writing
- Wide selection of datasets, tasks, models and baselines for experimental evaluation
- relevant topic

Weaknesses:
- While the experimental setup is extensive, I see potential red flags that I would ask the authors to address (detailed below)
- Baselines: A common approach to improved server-side averaging is to use FedAdam (or other higher-moment optimizers) at the server (https://arxiv.org/abs/2003.00295). I would like to understand why the authors did not compare against these algorithms

---

### Review · Reviewer_LveH · 2023-11-01

**Summary Of Contributions:**

This paper studies the problem of learning an unfair model in federated learning, where the trained model may unfairly advantage or disadvantage some of the devices. The authors propose AdaFed which adaptively finds an updating direction for the server along
which (i) all the clients’ loss functions are decreasing; and (ii) the loss functions for the clients with larger values decrease with a higher rate. This results in a method which better balances the training across multiple devices which is effective on a suite of federated datasets in improving fairness.

**Audience:**

Yes

**Claims And Evidence:**

No

**Requested Changes:**

see weakness points 1-4

**Strengths And Weaknesses:**

Strengths:
1. The paper is solid and well motivated, and generally well written.
2. The method is very well presented and analyzed, with theoretical results supporting the main claims.
3. Experiments are extensive over many FL datasets and settings, results are consistently better than baselines.

Weaknesses:
1. Although the authors have acknowledged the lack of standard definition of fairness, I am wondering if there is another more suitable term - fairness is commonly used to refer to social biases or performance disparities across different individuals or groups in the ML literature. Furthermore, the authors have not really analyzed why unfairness can occur - is it mostly due to imbalanced client dataset sizes? out of distribution clients, clients with data from a different modality? Analyzing these reasons seems important to built solutions.
2. Quite a few missing baselines, in particular FedMGDA which is directly built-upon for motivation is a missing baseline in Tables 2,3,4. TERM and Ditto are also missing in some of the Tables.
3. Section 7.5 Analysis of Results can be significantly expanded. It would be good if the main theorems regarding convergence and scaling are analyzed empirically here as well. Another experiment would to more closely analyze which populations/devices the model found to be unfair and ended up scaling to improve fairness. Any insights here for how to better design FL distributions/study fairness/improve modeling?
4. Not a critical change, but the paper would be much more impactful if there were some real-world FL experiments with actual data distributions that exhibit unfairness when trained normally, such as certain hospitals, demographic groups, etc.

---

### Decision · Action_Editor_2Jxp · 2023-12-22

**Recommendation:** Accept as is

**Comment:**

The reviewers unanimously commended the exposition of the paper citing that it is clear and coherent. The reviewers also found the claims to be correct. The problem considered by the paper is also of potential interest to the federated learning community. Some suggestions made by reviewers were partially addressed but they are not critical.

**Audience:**

The paper should be of interest federated learning community.

**Claims And Evidence:**

The reviewers found the claims made in the submission to be accurate, convincing and clear. The reviewers commended the paper on its exposition and details.